# MindGYM: What Matters in Question Synthesis for Thinking-Centric Fine-Tuning?

**Zhe Xu**[1*], **Daoyuan Chen**[2*], **Zhenqing Ling**[1*], **Yaliang Li**[2], **Ying Shen**[1,3†]

[1]Sun Yat-sen University, [2]Alibaba Group, [3]FSIETP

## Abstract

Large foundation models face challenges in acquiring transferable, structured thinking abilities, especially when supervised with rigid templates or crowd-annotated instruction datasets. Unlike prior approaches, we focus on a thinking-centric data synthesis paradigm that enables models to evolve through self-generated, cognitively guided data. We propose MINDGYM, a structured and scalable framework for question synthesis, composed of: (1) Cognitive Thinking Process Injection, which infuses high-level reasoning objectives to shape the model's synthesis behavior; (2) Seed Single-Hop Question Synthesis, generating atomic questions from diverse semantic types to encourage broader thinking; and (3) Challenging Multi-Hop QA Synthesis, composing more complex multi-hop questions based on QA seeds for deeper reasoning. Detailed analysis shows that synthetic data generated by our method achieves 16.7% higher average quality and 67.91% lower quality variance compared to baseline sources, highlighting that both high-quality and self-contained data are essential for effective, thinking-oriented finetuning. MINDGYM improves performance on six reasoning benchmarks, achieving gains of up to 16% on MathVision using only 400 data samples, and generalizable improvements across different model sizes and architectures. MINDGYM underscores the viability of self-challenging mechanisms in refining large model capabilities while minimizing human intervention and resource demands. Code and data are released to promote data-centric research into self-evolving foundation models driven by their internal reasoning capabilities.

## 1 Introduction

Large foundation models have emerged as key assistants for tasks requiring complex understanding across tasks and modalities [3, 30]. However, enabling robust performance with transferable and efficient thinking abilities remains challenging [4, 6]. Manually curated instruction datasets like OK-VQA [24] and ScienceQA [22] are labor-intensive to scale, while self-supervised synthetic methods such as MMInstruct [20] and MMEvol [23] suffer from limited generalization across modality and task types, often failing to produce logically consistent or cognitively diverse data. Meanwhile, reasoning enhancement methods—such as reinforcement learning (RL) [1] or iterative prompting [17]—incur prohibitive computational costs, limiting their practicality.

To tackle these limitations, we introduce MINDGYM, a **thinking-centric data synthesis framework** aimed at enhancing the cognitive capacity of large models. Rather than relying on task-specific templates or crowd-sourced samples, our approach embeds structured thinking traits into the synthesis process — enabling models to self-generate data that target their cognitive bottlenecks. The proposed framework consists of three key components illustrating in Figure 1:

---

*Equal contribution.

†Corresponding author.

[3]FSIETP: Guangdong Provincial Key Laboratory of Fire Science and Intelligent Emergency Technology.

1. **Cognitive Thinking Injection**: We infuse structured thinking priors — such as breadth (cross-topic linkage), depth (multi-step deduction), and progression (difficulty scaling) — into the prompt design, guiding generation toward cognitively rich and pedagogically effective samples.

2. **Seed Single-Hop Question Synthesis**: We synthesize a set of semantically grounded, single-hop QA primitives that span diverse categories including arithmetic, logic, ethics, and causality. These act as composable base units for constructing multi-hop cognitive challenges.

3. **Challenging Multi-Hop QA Synthesis**: By composing seed questions using thinking-centric operators such as *Bridging*, *Comparison*, and *Temporal*, we generate challenging multi-hop questions that demand higher-order inference and cross-domain understanding.

We validate MINDGYM on six reasoning-intensive benchmarks across multiple vision-language models (VLMs) architectures. As shown in Table 1, with only 400 synthetic samples, our method yields significant gains: for example, our method boosts performance of Qwen2.5-VL-7B [3] by over 16% on MathVision-Mini. These improvements are consistent across models of varying scales and architectures, demonstrating the generality of our approach. Further analysis via DATA-JUICER [4] reveals that on Qwen2.5-VL-32B [3], our synthetic data achieves a +16.7% average quality improvement over baselines and a 67.91% reduction in quality variance. This highlights an important insight: not only does cognitively guided synthesis yield higher-quality data, but *stability in quality*—as measured by low variance—is critical for consistent fine-tuning. Compared with CoT [38] and ToT [41] baselines, our multi-step cognitive reasoning framework demonstrates superiority. We further observe that Chinese-synthesized data outperform English and mixed-language variants. Moreover, beyond the high-quality textual data synthesis presented in the main page, we also explore MINDGYM's *scalability to multimodal settings* in Appendix C, demonstrating its effectiveness and potential to both VLMs and large language models (LLMs).

To summarize, our contributions are three-fold:

- We propose MINDGYM, a scalable, model-dependent self-synthesis framework for thinking-centric data, which injects structured cognitive priors into both single-hop and multi-hop question generation, encouraging both broad and deep reasoning.

- We present an in-depth data analysis, showing that reducing variance is as critical as improving mean quality for stable fine-tuning. The proposed method generates cognitively guided data that excels on both fronts.

- We demonstrate that self-challenging data evolution significantly enhances reasoning performance across six benchmarks, even under limited supervision, and generalizes well across different models. All code and data are released at https://github.com/modelscope/data-juicer/tree/MindGYM/ to shed light on further research and applications.

## 2 Related Works

**Instruction Data Construction for VLMs.** High-quality instruction data is pivotal for vision-language models (VLMs) to align with human intent [15, 5]. While early benchmarks like OK-VQA [24] and ScienceQA [22] rely on costly human annotation, recent work embraces self-instruction paradigms to scale data generation [37, 34, 40]. In the multimodal domain, MMInstruct [20] and MMEvol [23] extend this idea, but often suffer from modality inconsistency, where visual inputs are weakly grounded to textual outputs. These limitations stem from over-reliance on LLMs without structured visual reasoning. Our work departs from these trends by embedding cognitively structured priors into prompt design, generating semantically aligned, cross-modal instruction data.

**Thinking-Centric Question Synthesis** Synthetic data generation has long been a tool for improving model reasoning. In language-only domains, question synthesis approaches have emphasized structure, difficulty scaling, and multi-step deduction, particularly in math [16] or logic domains [18, 32]. However, most existing methods either depend on rigid templates or domain-specific structures, which constrains their generalizability. Multimodal question synthesis introduces further complexity due to the need for cross-modal reasoning and abstract concept grounding. Methods like Visual Program Induction [10] or GRILL [27] attempt structured generation but often produce shallow or repetitive questions. Moreover, many pipelines prioritize syntactic fluency over cognitive depth, failing to elicit the higher-order thinking needed for robust generalization. This motivates the need for synthesis frameworks that directly embed cognitive traits—such as reasoning breadth, depth, and progression—into the data itself.

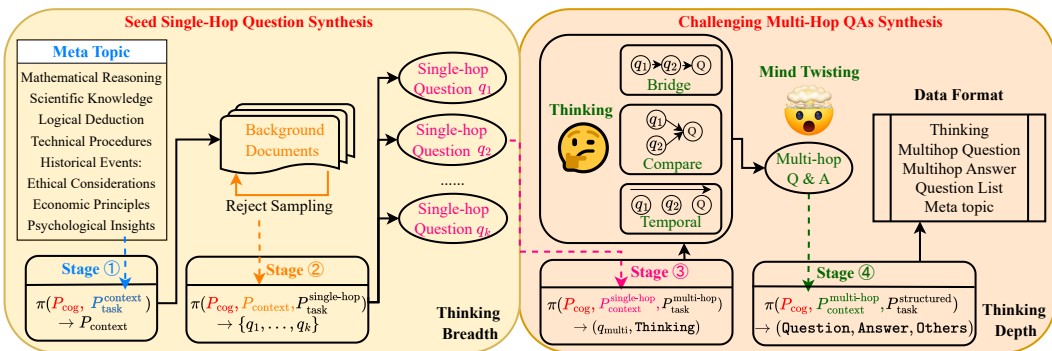

Figure 1: The proposed MINDGYM framework incorporates a cognitively guided data synthesis pipeline with four stages: ① **Context Generation**, ② **Single-Hop Question Synthesis**, ③ **Multi-Hop Composition with Thinking Trace**, and ④ **Structured Extraction**. Starting from a meta-topic and guided by a shared cognitive objective, the model iteratively builds background context, atomic reasoning steps, composite questions with interpretable thinking traces, and final structured QA samples for downstream use. Color-coded regions and arrows illustrate the hierarchical progression from simple reasoning to advanced problem-solving and emphasizing conceptual depth.

**Self-Improving Reasoning with Limited Supervision**   Self-improvement paradigms for reasoning have emerged to reduce human intervention and labeling costs. Reinforcement learning–based methods (e.g., RL4F [1]) and iterative self-refinement [17] fine-tune models through repeated exploration and feedback loops but require significant computational resources. Lightweight techniques like Chain-of-Thought prompting [38] and Tree-of-Thoughts [41] improve test-time reasoning but often rely on long sequences or search-time strategies, which are inefficient for deployment. In contrast, data-centric self-improvement strategies aim to modify the training signal itself rather than the inference process. MINDGYM fits into this paradigm by synthesizing cognitively aligned, reasoning-targeted samples that directly support robust and efficient learning, without the need for test-time decoding tricks or expensive policy optimization.

## 3   The Proposed MINDGYM

In this work, we aim to embed advanced thinking capabilities into large models through self-synthesized fine-tuning datasets. Prior research has shown that conventional single-hop question–answer pairs are often inadequate to stimulate thinking along both depth and breadth dimensions, especially for tasks that require multi-step deduction or complex reasoning. To address this, we introduce a progressive self-synthesis framework, outlined in Figure 1, operating through three interconnected components: (1) Cognitive Thinking Injection, where diverse human-like reasoning strategies are embedded into the synthesis process via structured prompts; (2) Seed Single-Hop Question Synthesis, where the model generates logically grounded, cognitively-aware atomic questions; and (3) Challenging Multi-Hop QA Synthesis, where these atomic units are composed into complex, multi-step questions with explicit reasoning trajectories.

This framework enhances generalization by exposing models to diverse cognitive challenges while injecting interpretable thinking patterns that mirror human-like problem-solving dynamics into the model parameters. Detailed designs and implementations are provided in subsequent subsections.

### 3.1   Cognitive Thinking Injection

**Cognitive-Aware Synthesis Templates.**   To guide models toward human-like thinking, we design a cognitive-aware synthesis prompt composed of three elements: cognitive setting $P_{\text{cog}}$, background conditions $P_{\text{context}}$, and synthesis objectives $P_{\text{task}}$. Among them, $P_{\text{context}}$ and $P_{\text{task}}$ vary across different synthesis scenarios (e.g., single-hop, multi-hop, text-only, or multimodal), while $P_{\text{cog}}$ remains consistent to ensure persistent cognitive grounding throughout all generation stages[4].

---

[4]See Appendix A.2 for the full prompt templates.

We propose a four-stage cognitive thinking injection framework that incrementally builds structured reasoning traces using a unified generation model $\pi$, guided by $P_{\text{cog}}$ and stage-specific $P_{\text{task}}$, with Stage 1 and Stage 2 described in Section 3.2, and Stage 3 and Stage 4 in Section 3.3:

- **Stage 1: Background Context Generation.** Given a meta-topic, the model generates a background passage $P_{\text{context}}$ using $P_{\text{cog}}$ and a context prompt $P_{\text{task}}^{\text{context}}$: $\pi(P_{\text{cog}}, P_{\text{task}}^{\text{context}}) \rightarrow P_{\text{context}}$.

- **Stage 2: Single-Hop Question Generation.** Conditioned on $P_{\text{cog}}$, $P_{\text{context}}$, and $P_{\text{task}}^{\text{single-hop}}$, the model generates $k$ logically independent seed questions: $\pi(P_{\text{cog}}, P_{\text{context}}, P_{\text{task}}^{\text{single-hop}}) \rightarrow \{q_1, \ldots, q_k\}$.

- **Stage 3: Cognitive Composition with Adaptive Types.** Using the single-hop questions and a multi-hop prompt, the model composes a complex multi-hop question and a corresponding reasoning trace: $\pi(P_{\text{cog}}, P_{\text{context}}^{\text{single-hop}}, P_{\text{task}}^{\text{multi-hop}}) \rightarrow (q_{\text{multi}}, \texttt{Thinking})$.

- **Stage 4: Structured Extraction.** Given the multi-hop results and a structured prompt, $\pi$ produces the final schema-aligned data: $\pi(P_{\text{cog}}, P_{\text{context}}^{\text{multi-hop}}, P_{\text{task}}^{\text{structured}}) \rightarrow (\texttt{Question}, \texttt{Answer}, \texttt{Others})$.

**Adaptive Stream of Consciousness.** To implement $P_{\text{cog}}$, we adapt the Thinking-Claude protocol [43], structuring it as a flexible suite of cognitive operations (e.g., hypothesis formulation and testing, multi-angle interpretation, counterfactual reasoning, and self-verification). These thinking elements are modular and reusable, enabling the model to emulate layered, human-like thinking. Across stages, $\pi$ is prompted to verbalize and externalize its internal thinking steps, thereby producing interpretable structured traces that reveal how the model navigates complex problem spaces.

This cognitively guided synthesis approach enhances generalization by exposing the model to diverse reasoning challenges while injecting structured and interpretable thinking into training data. The resulting dataset contains rich cognitive signals that are later leveraged during fine-tuning (see Section 3.4). Empirical evidence for the efficacy of $P_{\text{cog}}$ is presented in Section 4.4.

### 3.2 Seed Single-Hop Question Synthesis

**Cognitively Grounded Meta-Topics.** To ensure reasoning diversity and depth, we construct eight orthogonal *meta-topics* derived from established cognitive theories, including unified cognitive architectures [26] and dual-representation models [33]. These meta-topics span four core reasoning dimensions: **quantitative**, **causal**, **temporal**, and **social-ethical**, offering a comprehensive blueprint for cognitively rich question generation. Detailed meta-topics are provided in Appendix A.2.

**Stage 1: Background Context Generation.** Given a sampled meta-topic and its brief description, the model is prompted with a cognitive grounding prompt $P_{\text{cog}}$ and a context-specific instruction $P_{\text{task}}^{\text{context}}$ to self-generate a short but semantically rich background passage $P_{\text{context}}$. This passage serves as a reference for subsequent single-hop question generation.

**Stage 2: Single-Hop Question Generation.** Conditioned on the generated background passage $P_{\text{context}}$, the cognitive intent $P_{\text{cog}}$, and a task-specific instruction $P_{\text{task}}^{\text{single-hop}}$, the model generates a small batch (up to $k = 5$) of logically grounded, atomic single-hop questions $\{q_1, q_2, \ldots, q_k\}$. These questions are designed to probe individual reasoning steps based on the passage and are explicitly treated as intermediate components for later multi-hop composition in Section 3.3.

**Reject Sampling for Diversity.** To enhance diversity in seed question generation, MINDGYM further applies semantic vectorization to textual components of data in the synthesis pool [19]. We use an empirically determined cosine similarity threshold to evaluate the overlap between newly generated and existing samples. If similarity exceeds the threshold, the system initiates a regeneration cycle to eliminate semantically redundant content, continuing until the seed question pool reaches the target size $N$. This reject sampling mechanism widens the breadth of reasoning in synthetic data.

Unlike prior works that narrowly focus on limited task domains [2, 14] (e.g., arithmetic, logic puzzles), MINDGYM **emphasizes** (1) internalized background grounding, (2) cognitively structured topic guidance, and (3) semantic diversity enforcement. In Section 4.4, we empirically validate MINDGYM robustness and effectiveness across varied synthesis sources, including model's internal knowledge and multimodal contexts.

### 3.3 Challenging Multi-Hop QA Synthesis

**Stage 3: Cognitive Composition with Adaptive Types.** Given the seed questions $\{q_i^j\}_{j=1}^k$ from Section 3.2, we treat them as updated context $P_{\text{context}}^{\text{single}}$ for this phase. The model $\pi$ is now prompted with the triplet $(P_{\text{cog}}, P_{\text{context}}^{\text{single}}, P_{\text{task}}^{\text{multi-hop}})$ to synthesize a more complex, cognitively grounded question $Q_i$. The task instruction $P_{\text{task}}^{\text{multi-hop}}$ specifies an abstract reasoning requirement (e.g., comparing entities, bridging concepts, or analyzing temporal relations), which guides the model to self-compose from prior seeds. Details of composition types are provided in Appendix A.

**Stage 4: Structured Extraction.** To support modular training and interpretation, we encourage $\pi$ to format its output in three clearly separated blocks: the final multi-hop question $Q_i$, its answer $A_i$, and the structured rationale $T_i$. These thinking traces serve as valuable training signals for reasoning-intensive QA tasks, which is used for training in Section 3.4).

### 3.4 Usage and Discussion

**Training with the Dataset.** In this section, we explore how to effectively utilize our cognitively annotated dataset $\{(Q_i, A_i, T_i)\}_{i \in N}$ by organizing the training process into a gradually evolving pathway that transitions from explicit external guidance ($T_i$) to self-contained reasoning. The process begins with *guided answering*, where the model receives both the question and the reasoning trace ($Q_i$, $T_i \rightarrow A_i$) to encourage answer generation aligned with structured thought. It then advances to *reason reconstruction*, where the model is given the question and answer and learns to infer the underlying rationale ($Q_i, A_i \rightarrow T_i$), reinforcing logical coherence. Next comes *paired reasoning*, which requires the model to jointly generate both the answer and rationale from the question alone ($Q_i \rightarrow A_i$, $T_i$), simulating independent reasoning under minimal supervision. Finally, in the *autonomous solving* phase, the model learns to directly produce the answer from the question ($Q_i \rightarrow A_i$), fully internalizing the reasoning process. This learning pathway reflects the natural progression of human cognitive development—starting from external support and moving toward self-directed problem solving. As demonstrated in Section 4.4, this structured progression leads to significantly better performance compared to training with uniform supervision.

**Extensibility of the Dataset.** In addition to our primary focus on cognitively annotated textual data, we conduct preliminary experiments on extending our synthesis framework to the multimodal setting. As described in Appendix C, we leverage VLMs to generate multi-hop QA samples grounded in visual context. Specifically, we use OK-VQA [24] and ScienceQA [22] as anchor image sources to guide the synthesis process. The generated results summarized in Appendix C.3 show encouraging quality and reasoning depth, suggesting the potential of our framework beyond the text-only domain. Nonetheless, multimodal synthesis with VLMs remains an open challenge. Current reliance on static image datasets as anchors limits the diversity and scalability of visual reasoning. Looking forward, we plan to (1) incorporate richer and more diverse image datasets to better stimulate cognitive engagement, and (2) further explore fully generative pipelines where VLMs synthesize both the image and the associated reasoning task, enabling models to improve through self-generated multimodal experiences—a key step toward the self-evolution of models.

## 4 Experiments

To comprehensively evaluate the performance of our proposed synthetic data method, we conduct extensive experimental validation on multiple representative evaluation sets. Full descriptions of datasets, benchmarks, implementation details, and baselines are provided in Appendix B.

### 4.1 Experimental Settings

**Models & Implementation.** The data synthesis in MINDGYM leverages VLMs' inherent capabilities for long-context understanding and reasoning. We adopted four vision language models to verify the generality of our data synthesis method. These models are selected to cover both scaling within a single model family (Qwen2.5-VL 7B and 32B) and variation across different models (Qwen and InternVL series). For all experiments in the main text, we focus on text-only data synthesis, where only the LLM-layers are updated during training, while all vision-related layers remain frozen.

Table 1: Performance of different models on various evaluation benchmarks. The number following the dataset name indicates its size in terms of the number of samples used for training.

| Models | Dataset (# of Samples) | Text Eval | | | Multimodal Eval | | | OVERALL |
|---|---|---|---|---|---|---|---|---|
| | | GSM8K | Math | GPQA | MMStar | MathVista | MathVision | |
| Qwen2.5-VL-7B | raw | 83.62 | 67.60 | 31.83 | 64.00 | 69.30 | 24.67 | 56.84 |
| | Openo1-sft (400) [35] | **84.31** | **69.00** | 29.70 | 63.87 | 69.70 | 23.66 | 57.04 (+0.20) |
| | Openo1-sft (4k) [35] | 77.94 | 59.00 | 28.19 | 58.93 | 61.40 | 17.43 | 50.48 (-6.36) |
| | LIMO (817) [42] | 84.08 | 67.80 | 30.58 | 63.93 | 70.20 | 25.90 | 57.10 (+0.26) |
| | MMEvol-SciQA (106) [23] | 83.40 | 65.60 | 31.83 | 63.93 | 69.50 | 23.36 | 56.27 (-0.57) |
| | MMEvol-DvQA (4k) [23] | 83.85 | 65.80 | **33.08** | 62.93 | 67.40 | 24.01 | 56.18 (-0.66) |
| | **MINDGYM-Text (400)** | 84.08 | 68.4 | 31.33 | **64.33** | **70.30** | **28.62** | **57.84 (+1.00)** |
| Qwen2.5-VL-32B | raw | 95.15 | 81.80 | 47.98 | **69.60** | 73.40 | 37.50 | 67.58 |
| | Openo1-sft (400) [35] | 95.38 | 81.20 | **50.51** | 68.80 | 73.40 | 36.18 | 67.58 (+0.00) |
| | Openo1-sft (4k) [35] | 95.68 | 79.80 | 38.00 | 68.27 | 72.20 | 34.54 | 64.75 (-2.83) |
| | LIMO (817) [42] | **95.83** | 80.80 | 41.92 | 69.33 | 71.80 | 37.83 | 66.25 (-1.33) |
| | MMEvol-SciQA (106) [23] | 95.60 | 81.00 | 42.42 | 69.00 | **73.80** | 36.84 | 66.44 (-1.14) |
| | MMEvol-DvQA (4k) [23] | 92.72 | 80.60 | 38.38 | 69.20 | 72.60 | 39.14 | 65.44 (-2.14) |
| | **MINDGYM-Text (400)** | 95.53 | **82.00** | 48.48 | 69.10 | 72.80 | **40.46** | **68.06 (+0.48)** |
| InternVL-8B | raw | 86.50 | 76.30 | 33.84 | 69.00 | 73.20 | 33.22 | 62.46 |
| | Openo1-sft (400) [35] | **89.39** | 77.20 | 43.43 | 69.00 | **73.60** | 32.57 | 64.20 (+1.74) |
| | Openo1-sft (4k) [35] | 88.55 | 78.00 | 42.42 | 68.87 | 72.60 | 31.91 | 63.72 (+1.26) |
| | LIMO (817) [42] | 89.16 | 76.40 | 42.42 | 69.00 | 73.40 | 32.57 | 63.82 (+1.36) |
| | MMEvol-SciQA (106) [23] | 88.40 | 77.80 | **46.97** | 69.13 | 72.80 | 33.88 | 64.83 (+2.37) |
| | MMEvol-DvQA (4k) [23] | 88.55 | 74.60 | 40.40 | 68.47 | 71.60 | 32.24 | 62.65 (+0.19) |
| | **MINDGYM-Text (400)** | 88.63 | **78.20** | 45.45 | 69.27 | 72.80 | **35.53** | **64.95 (+2.49)** |
| InternVL-38B | raw | 89.16 | 72.60 | 47.47 | 72.40 | 72.80 | 36.51 | 65.16 |
| | Openo1-sft (400) [35] | 89.46 | 76.4 | 48.99 | 72.27 | 72.00 | 35.53 | 65.77 (+0.61) |
| | Openo1-sft (4k) [35] | 89.46 | **79.20** | 44.95 | **72.47** | 72.90 | 35.53 | 65.75 (+0.59) |
| | LIMO (817) [42] | 89.16 | 72.60 | 47.47 | 72.40 | 72.80 | 36.51 | 65.16 (+0.00) |
| | MMEvol-SciQA (106) [23] | 89.46 | 75.00 | 46.46 | 72.40 | **73.20** | **38.49** | 65.83 (+0.67) |
| | MMEvol-DvQA (4k) [23] | **89.92** | 74.00 | 46.46 | 70.93 | 71.70 | 30.92 | 63.99 (-1.17) |
| | **MINDGYM-Text (400)** | 89.46 | 74.80 | **50.00** | 72.40 | 72.90 | 37.83 | **66.23 (+1.07)** |

**Datasets.** We focus exclusively on synthetic textual data generation in the main body of our work, relying solely on self-contained knowledge extraction from the base model without external data sources. To verify the robustness and generality of our synthesis method, we employ the aforementioned models to perform self-consistent data generation. Each model independently synthesizes 400 Chinese samples. The choice of Chinese is driven by its linguistic complexity and higher information density, making the synthesis task more challenging.

**Benchmarks.** To systematically assess MINDGYM on VLMs across reasoning modalities, we incorporate: (1) **Text-based evaluation** sets comprise GSM8K [8], MATH [12], and GPQA [31]; (2) **Multimodal evaluation** sets include MMStar [7], MathVista-Mini [21], and MathVision-Mini [36]. The OVERALL quantifies comprehensive performance spanning both text and multimodal domains.

**Baselines.** As MINDGYM generates self-contained adversarial training data without external sources, direct comparison with conventional data synthesis methods proves infeasible. We therefore select SOTA reasoning-oriented dataset works from two categories: (1) Text-based reasoning with LIMO [42], 817 curated logical chains, and OPEN-O1 [35], an SFT dataset for CoT activation as fundamental text reasoning baselines, and (2) Multimodal reasoning using MMEVOL's [23] core subsets SCIENCEQA and DVQA. All baseline datasets are used in accordance with their original supervised fine-tuning setting. For fair comparison, we apply identical training protocols and hyperparameter configurations across all methods.

## 4.2 Main Results

To validate the efficacy of our methodology, we systematically evaluate the model across multiple benchmarks as documented in Table 1, with key insights summarized:

**Superiority across Different Tasks.** MINDGYM demonstrates substantial performance advantages through comprehensive average metric evaluation. Notably, on the Qwen2.5-VL-7B, MINDGYM achieves the OVERALL of **57.84**, outperforming all competing baselines. It achieves the highest

MathVision score (**28.62**), exceeding the Raw model by **16%** and the strongest baseline LIMO by **2.72** points. An interesting phenomenon emerges among baselines that OPEN-O1 (4K) suffers severe degradation with **29.3%** MathVision collapse despite employing a large number of training samples, exposing the vulnerability of conventional scaling approaches.

**Superiority on Different Models.** MINDGYM consistently demonstrates strong robustness across both diverse models and varying parameter scales. On the Qwen2.5-VL series, MINDGYM improves the Overall performance from 57.84 to 64.33 on the 7B model, a relative gain of **+0.74%–7.36%**, and from 68.06 to 68.06 on the 32B model, reflecting a more modest but stable **+0.48%–3.31%** improvement. Similarly, for InternVL3, the Overall score rises from 64.95 to 64.95 on the 14B model **(+0.12%–2.30%)**, and from 66.23 to 66.23 on the 38B model **(+0.40%–2.24%)**. These gains remain consistently positive across all four backbones, even as baseline capabilities increase with scale. Notably, the relative improvements are more substantial at smaller model sizes (e.g., Qwen2.5-VL-7B), suggesting MINDGYM is particularly effective in lower-resource or less capable models. This trend underscores the method's architecture-agnostic and scaling-friendly design, making it highly transferable across a wide range of multimodal foundation models.

**Data Efficiency.** Despite using only **400** synthetic training samples, MINDGYM achieves performance that not only matches but exceeds baselines trained with **10×** more data. This includes outperforming OPEN-O1 (4K) in both textual and multimodal tasks. On MathVision, MINDGYM reaches **28.62** while OPEN-O1 (4K) collapses to **17.43**. This stark contrast underscores the high quality and utility of the data synthesized by MINDGYM.

**Cross-modal Enhancement.** Despite containing only 400 synthetic examples in pure-text format, MINDGYM significantly enhances both textual and cross-modal reasoning capabilities. As shown in Table 1, our method improves the OVERALL performance not only on language-only tasks, but also on multimodal benchmarks (e.g., from **62.46** to **64.95** on InternVL3-14B and **65.16** to **66.23** on InternVL3-38B). These consistent improvements are also observed in vision-intensive benchmarks like MathVista and MathVision, even though no vision data was used in fine-tuning. This demonstrates that our synthetic dataset effectively strengthens the model's intrinsic thinking capability, enabling better generalization across modalities. Such cross-modal enhancement, achieved through text-only supervision, highlights the thinking-centered nature of our data and its ability to propagate improvements to both language and vision domains.

**Summary.** These results collectively demonstrate that our method effectively tackles three key challenges in multimodal instruction tuning: **data inefficiency**, **modality transferability**, and **scalability across different models and sizes**. First, MINDGYM achieves consistent and significant gains across all evaluation metrics with only 400 synthetic Chinese text-only samples, outperforming baselines trained with up to 10× more data, highlighting its exceptional data efficiency. Second, the improvements span both textual and vision-intensive benchmarks, even without visual supervision, indicating that our thinking-centric text data successfully enhances intrinsic thinking that transfers across modalities. Third, our method remains robust and effective across a range of backbones (Qwen2.5-VL and InternVL3) and model sizes (7B–38B), making it highly scalable and architecture-agnostic. These findings validate that our approach addresses the limitations of prior methods in terms of *modality inconsistency*, *generalization*, and *computational efficiency* (as outlined in Section 2).

### 4.3 Data analysis

To comprehensively assess the quality of our MINDGYM synthetic dataset, we adopt DATA-JUICER[4][5], a modular data-centric analysis toolkit that provides LLM-guided operators for probing data across various dimensions. Specifically, we compare MINDGYM against several baselines (e.g., LIMO, Open-O1, MMEvol) using five diagnostic filters: (1) **quality**, which estimates the overall textual quality based on LLM judgment; (2) **action**, counting the number of action verbs as a proxy for instruction richness; (3) **dependency**, which penalizes the presence of non-independent noun phrases based on syntactic trees; (4) **token**, evaluating token-length consistency; and (5) **length**, which considers raw text length variability. Table 2 summarizes the full results across multiple model series and checkpoints. Our key findings are elaborated below.

---

[5]`https://github.com/modelscope/data-juicer`

Table 2: Data-Juicer analysis results across different models.

| Model | Dataset | quality-mean ↑ | quality-std ↓ | action ↑ | dependency ↑ | token | length |
|-------|---------|--------------|-------------|----------|--------------|-------|--------|
| baseline | Openo1-sft (400) [35] | 0.96 | 0.091 | 8.05 | 2.00 | 77 | 274 |
| | Openo1-sft (4k) [35] | 0.70 | 0.10 | 8.78 | 2.02 | 83 | 284 |
| | LIMO (817) [42] | 0.87 | 0.10 | 7.71 | 2.02 | 101 | 322 |
| | MMEvol-SciQA (106) [23] | 0.91 | 0.082 | 2.31 | 1.85 | 13.73 | 67 |
| | MMEvol-DvQA (4k) [23] | 0.76 | 0.11 | 2.66 | 1.95 | 14.82 | 69 |
| Qwen2.5-VL-7B | MINDGYM-Text (ours) | 0.93 | 0.055 | 10.34 | 2.17 | 147 | 110 |
| Qwen2.5-VL-32B | MINDGYM-Text (ours) | 0.98 | 0.031 | 10.94 | 2.25 | 168 | 134 |
| InternVL-8B | MINDGYM-Text (ours) | 0.97 | 0.044 | 8.64 | 2.16 | 233 | 195 |
| InternVL-38B | MINDGYM-Text (ours) | 0.96 | 0.039 | 7.66 | 2.07 | 118 | 92 |

**Higher Quality.** We observe a strong positive correlation between the average quality score (as estimated by DATA-JUICER[4]) and downstream fine-tuning performance. In the baseline group, Open-O1(400) and MMEvol-SciQA achieve high quality scores (**0.96** and **0.91**, respectively), and correspondingly rank first and second (excluding MINDGYM) in Table 1, outperforming other baselines. Notably, Open-O1(400) not only obtains the highest quality score among baselines, but also shows consistent Overall score improvements across all models com-

Table 3: The results of win rates in terms of relative improvements on MINDGYM over baselines, using GPT-4 as a scorer.

| Datasets | DEPTH | BREADTH | AVG |
|----------|-------|---------|-----|
| **Raw** | 10.2% | 19.4% | 14.8% ↑ |
| **LIMO** | 1.63% | 23.8% | 12.7% ↑ |
| **Open-O1 (400)** | 1.88% | 23.7% | 12.8% ↑ |
| **MMEvol-DVQA** | 8.41% | 37.0% | 22.7% ↑ |
| **OVERALL** | 5.53% ↑ | 26.0% ↑ | 15.8% ↑ |

pared to the raw version. MINDGYM further surpass these baselines in quality: Qwen2.5-VL-7B reaches **0.93**, while Qwen2.5-VL-32B achieves **0.98**. These results suggest that larger LLMs not only produce better answers, but also generate higher-quality synthetic data. This trend holds across models and highlights the value of scaling for data generation.

**Lower Variance.** Beyond high average quality, we observe a key phenomenon: as the parameter scale of the LLM used for data synthesis increases, the variance of quality scores decreases. For instance, Qwen2.5-VL-32B and InternVL3-38B exhibit low standard deviations (**0.031** and **0.039**, respectively), in contrast to baselines like MMEvol-DvQA, which shows a much higher variance (**0.11**) and ranks among the worst-performing in Table 1. Even within the baseline group, Open-O1(400) and MMEvol-SciQA, both with relatively lower variances, also achieve stronger Overall scores, typically outperforming their raw counterparts.

Interestingly, although MMEvol-SciQA has slightly lower variance than Open-O1(400), its downstream performance lags behind, for example, showing negative OVERALL gain under Qwen2.5-VL-32B. This can be attributed to its lower mean quality score, reinforcing that both high average quality and low variance are critical to data effectiveness. The low-variance patterns observed in larger models suggest their superior ability not only to generate high-quality content but also to suppress outliers and noise—an essential property for stable and robust fine-tuning. These findings underscore a key insight: *consistency matters as much as mean quality* in building high-impact synthetic datasets.

**Richer Semantics.** We next analyze linguistic properties that reflect structural complexity and task grounding: the average number of **action verbs** and the count of **non-independent entities** derived from dependency trees. MINDGYM samples show significantly higher values in both metrics. For instance, Qwen2.5-VL-32B samples contain **14.14** action verbs and **2.14** non-independent entities on average, while LIMO and MMEvol-SciQA yield only **7.71/2.02** and **2.31/1.85**, respectively. Meanwhile, MINDGYM maintains a compact format, averaging **133–215** tokens and **92–110** characters per sample across models. This balance of rich semantics and concise form enhances model understanding while improving supervision efficiency and signal clarity. These results suggest that MINDGYM prompts are both procedurally rich and semantically dense, which facilitates stronger self-consistency in instruction following.

**GPT-based Thinking Quality Scoring.** We establish a two-dimensional evaluation framework leveraging GPT-4 as an expert scorer. The protocol assesses: (1) *thinking Depth* via derivation

complexity, and (2) *thinking Breadth* by solution diversity. Each metric operates under a standardized scoring system, with detailed specifications provided in Appendix A.2.5. Comparative evaluations against four baselines are conducted on the MathVision dataset as a benchmark. The experimental results documented in Table 3 demonstrate MINDGYM's consistent superiority across both evaluation axes, particularly achieving **15.8%** average improvements in thinking depth over the raw model and other competitive baselines.

## 4.4 Ablation studies

To validate the efficacy of our approach, we conduct systematic module-wise utility analysis through targeted ablation studies on Qwen2.5-VL-7B[3].

**Impact of Structured Cognitive Thinking.** We investigate the role of the **structured cognitive thinking (SC)** module, which models multi-step reasoning behaviors such as planning, divergent exploration, and selective consolidation. This module is implemented through our multi-step synthesis scheme, where question and answer generation are explicitly decomposed into reasoning stages. To contrast with this design, we evaluate two widely used reasoning paradigms—*Chain-of-Thought* (CoT) and *Tree-of-Thought* (ToT)—which directly generate multi-hop questions without intermediate reasoning supervision. As shown in Table 4, MINDGYM-TEXT achieves the highest average score of 57.84, outperforming both CoT (55.86) and ToT (55.92). This demonstrates that multi-step structured synthesis provides richer

Table 4: Ablation study results of removing *structured cognitive thinking* (*w/o SC*), utilizing English in data synthesis (*Syn-EN*), changing the order of our fine-tuning (*w/o OF*) steps and Relation Balanced (*with RB*), structured cognitive thinking with CoT and ToT synthes (*CoT* and *ToT*), complete results detailed in Appendix D.

| Datasets | MM-AVG | TEXT-AVG | AVG |
|---|---|---|---|
| Raw | 52.66 | 61.02 | 56.84 |
| MINDGYM-TEXT | 54.42 | 61.27 | 57.84 |
| MINDGYM-TEXT *w/o SC* | 52.72 | 60.00 | 56.36 ↓ |
| MINDGYM-TEXT *Syn-EN* | 52.77 | 60.30 | 56.53 ↓ |
| MINDGYM-TEXT *w/o OF* | 52.20 | 60.83 | 56.51 ↓ |
| MINDGYM-TEXT *with RB* | 52.20 | 60.83 | 56.51 ↓ |
| *CoT* | 59.74 | 51.98 | 55.86 ↓ |
| *ToT* | 59.78 | 52.07 | 55.92 ↓ |

reasoning signals than direct single-pass generation. Moreover, removing SC (MINDGYM-TEXT *w/o SC*) results in a performance drop to 56.36, confirming that structured cognitive guidance itself contributes to the improvement. Overall, these results suggest that **multi-step, cognition-guided synthesis is more effective than direct CoT or ToT prompting** for constructing high-quality reasoning data.

**Effect of Data Quality and Filtering.** We further analyze the **quality and impact of our data** through a controlled qualitative and filtering study. The dataset is evaluated by **GPT-4o** along four dimensions: logical consistency, linguistic clarity, factual correctness, and hallucination tendency. As shown in Table 5, over 89% of samples are judged factually

Table 5: GPT-4o-based qualitative evaluation of synthesized data.

| | logically flawed | syntactically ambiguous | correctness | hallucination |
|---|---|---|---|---|
| false | 328 | 394 | 357 | 372 |
| true | 71 | 5 | 42 | 27 |

correct, 93% non-hallucinatory, and fewer than 1% syntactically ambiguous, demonstrating the overall linguistic and logical soundness of the MINDGYM synthesis process.

To assess whether further cleaning benefits model training, we remove samples flagged as logically flawed or hallucinatory by GPT-4o and retrain the model on this "cleaned" subset. As presented in Table 6, the cleaned data lead to slightly lower performance compared to the full dataset (57.49 vs. 57.84). This finding indicates that the effectiveness

Table 6: Impact of filtering on model performance.

| Model | text-avg | MM-avg | Overall Avg |
|---|---|---|---|
| MINDGYM (full) | 61.27 | 54.42 | **57.84** |
| MINDGYM (clean) | 61.08 | 53.89 | 57.49 |

of MINDGYM does not solely stem from factual accuracy, but rather from the **rich reasoning trajectories** embedded in the synthesized data. Even imperfect samples often encode meaningful intermediate reasoning steps—such as decomposition, hypothesis formation, and evidence evaluation—that contribute to more generalizable reasoning behavior. These results suggest that, for **thinking-centric instruction tuning**, maintaining reasoning diversity and cognitive structure is more beneficial than enforcing strict factual precision.

**Effect of Multimodal Synthesis and Vision Backbone Freezing.** We assess the influence of multimodal synthesis and vision backbone freezing. As shown in Table 7, the default Chinese text-only setting with a frozen encoder achieves the best overall average of 57.84. Introducing Flux-based image generation slightly reduces performance to 56.80, while unfreezing the vision backbone leads to a further drop to 56.35. These results indicate that additional multimodal synthesis and vision unfreezing introduce minor instability without clear benefit, validating our design choice of a text-dominant and frozen-encoder configuration for stable reasoning performance. For more information about Flux dataset synthesis, see the Appendix A.2.6.

Table 7: Ablation on multimodal synthesis and vision backbone freezing. "CN" = Chinese text-only synthesis; "Flux" = additional text-to-image generation using Flux; "Frozen/Unfrozen" indicates whether the vision encoder is trainable.

| Setting | text-avg | MM-avg | Avg |
| --- | --- | --- | --- |
| CN (text only, Frozen) | 61.27 | 54.42 | **57.84** |
| Flux (image gen., Frozen) | 60.91 | 52.68 | 56.80 |
| CN (text only, Unfrozen) | 60.40 | 52.30 | 56.35 |

**Linguistic Difference.** Based on the selected Qwen [3], we examine the impact of synthesis language using three variants of MINDGYM data: Chinese (CN), mixed Chinese–English (MIX; 50% Chinese and 50% English), and English (EN), as shown in Table 8. The Chinese-only configuration achieves the highest overall performance with an average score of 57.84 and is adopted as our default setting. In contrast, using English for synthesis leads to a noticeable performance drop of 1.34 points on average, while the mixed-language variant (MIX) also performs slightly worse than CN. These findings indicate that Chinese synthesis yields more effective reasoning supervision, likely benefiting from its higher information density and compact semantic structure, which align better with the thinking-centric nature of our data construction.

Table 8: Cross-lingual comparison of MINDGYM data synthesis.

| Setting | text-avg | MM-avg | Avg |
| --- | --- | --- | --- |
| CN (Chinese only) | 61.27 | 54.42 | **57.84** |
| MIX (CN + EN) | 61.16 | 53.51 | 57.34 |
| EN (English only) | 60.30 | 52.77 | 56.50 |

**The Usage of Dataset.** To verify the effectiveness of our fine-tuning module, we mix and shuffle the first three phase and train the LLM on text data, subsequently using the data of *autonomous solving phase* to constrain the answers' generation pattern. This experimental setup allows us to systematically examine whether disrupting the fine-tuning order impacts model performance. The experimental results shown in Table 4 under the *w/o OF* condition reveal a performance degradation of **1.33** points, highlighting the critical impact of the our utilized order on the synthesis data.

**Relationship Distribution Balanced.** To verify the effectiveness of allowing the model to autonomously select relationship categories, we balance the number of entries generated for each category, totaling 400 entries. The experimental results shown in Table 4 under the *with RB* condition indicate that enforcing category balance leads to a decline of **1.33** in overall performance. This finding suggests that the autonomous selection of relationships by LLM reflects its cognitive understanding.

## 5 Conclusion & Future Works

We present MINDGYM, a cognitively guided framework for synthesizing self-challenging vision-language data. By injecting high-level reasoning signals and composing multi-hop tasks, MINDGYM enables models to acquire structured thinking with minimal data and computation. Our synthetic data achieves higher quality and lower variance, leading to strong gains across reasoning benchmarks.

Future directions include extending cognitive relationships to dynamic visual scenarios [45]; exploring the application of MINDGYM in specific domains like medicine [25] and finance; developing adaptive scoring operators [4] for question complexity; and integrating agents [11] for data correctness verification in generalized RL environments.

## Acknowledgement

This research was supported by the Key-Area Research and Development Program of Guangdong Province (Grant No. 2024B1111060004) and in part by the New Generation Artificial Intelligence National Science and Technology Major Project (Grant No. 2025ZD0123003).

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

# Appendix

**Table of Contents**

# A  Implementation Details of MINDGYM

## A.1  Detailed Description of Meta-topics

Unlike prior works focused on limited domain combinations [2], our approach emphasizes broad cognitive coverage:

- *Mathematical Reasoning*: Solving quantitative problems using numbers and formulas.
- *Scientific Knowledge*: Understanding natural laws and scientific principles.
- *Logical Deduction*: Forming conclusions through logical progression from premises.
- *Technical Procedures*: Following step-by-step instructions for practical tasks.
- *Historical Events*: Analyzing past events and their consequences.
- *Ethical Considerations*: Evaluating decisions through moral frameworks.
- *Economic Principles*: Studying resource allocation and valuation.
- *Psychological Insights*: Exploring human behavior and cognition.

For purely textual contexts, we define three canonical combination types:

- *Comparison*: Identifying contrasts or similarities across entities or events;
- *Bridging*: Linking information across distant knowledge pieces via intermediate inference;
- *Temporal*: Reasoning across sequential or causal timelines.

These types serve as explicit cues in $P_{\text{task}}^{\text{multi-hop}}$, guiding $\pi$ to adopt different reasoning schemata.

## A.2 Utilized Prompts

### A.2.1 Stream of Consciousness

The protocol introduce in Section 3.2 is based on Thinking Claude, which is an innovative prompt engineering framework designed to enhance the reasoning capabilities of Claude AI models through simulated human-like cognitive processes. This paradigm-shifting approach employs a multi-stage cognitive architecture featuring chain-of-thought prompting, self-reflective mechanisms, and structured problem decomposition. We incorporate it into MINDGYM's data synthesis process to enhance the inference performance of the model[6].

### A.2.2 Prompts for Background Generation

On the text side, due to the low quality of sub-question generation without foundational documents, we create background documents for sub-question generation through iterative polling based on eight cognitive reasoning themes. On the image side, since text-to-image generation models are still underdeveloped and produce low-quality results, we directly utilize existing reasoning and trial-based QA datasets.

---

**Prompt 1: Background (Text)**

Write a background document of approximately 150-200 words. The document should describe a scenario or context that includes interconnected details, suitable for reasoning tasks. The document should focus on the reasoning category: {category}.

The document should be rich enough to support multi-hop reasoning in Chinese.

---

### A.2.3 Prompts for Seed Single-Hop Question Synthesis

These two prompts are used to synthesize sub-problems, **Prompt 2** is used to synthesize text sub-problems, and **Prompt 3** is used to synthesize images sub-problems.

---

**Prompt 2: Single-Hop Question (Text)**

Based on the background document provided, generate up to 5 logically connected sub-questions. The relationship between these sub-questions should belong to a single category:

- **Bridging:** requiring connecting facts or pieces of information from the document.

- **Comparison:** involving comparing two or more elements described in the document.

- **Temporal:** requiring reasoning about the order or timing of events.

Clearly state the relationship category that links all sub-questions in Chinese:

---

**Prompt 3: Single-Hop Question (Image)**

Based on the provided image, original question, and original answer, generate up to 5 logically connected sub-questions. The relationship between these sub-questions should belong to one of the following categories:

- **Visual-Textual Alignment:** Requiring alignment between visual (e.g., images, charts) and textual information.

---

[6]https://github.com/richards199999/Thinking-Claude

- **Spatial Reasoning:** Involving spatial relationships or geometric layouts.

- **Causal Inference:** Requiring reasoning about cause-and-effect relationships.

- **Contextual Synthesis:** Requiring synthesis of information across multiple modalities (e.g., text, images, charts).

Clearly state the relationship category that links all sub-questions in Chinese:

### A.2.4 Prompts for Challenging Multi-Hop QA Synthesis

These two prompts are used to synthesize multi-hop problems, **Prompt 4** is used to synthesize multi-hop problems of text, and **Prompt 5** is used to synthesize multi-hop problems of images.

**Prompt 4: Multi-Hop QA (Text)**

Combine the sub-questions into a single, complex multi-hop question. The question should require reasoning across the sub-questions and synthesizing information from the background document. Then, provide a detailed answer to the multi-hop question, ensuring it is consistent with the background document and sub-questions.

Synthesize a multi-hop question and its answer based on the above sub-questions and background document in Chinese. Please start Qwen thinking and return the thinking process:

**Prompt 5: Multi-Hop QA (Image)**

Combine the sub-questions into a single, complex multi-hop question.

The question should require reasoning across the sub-questions and synthesizing information from the image and original question.

Then, provide a detailed answer to the multi-hop question, ensuring it is consistent with the image and sub-questions.

Synthesize a multi-hop question and its answer based on the above sub-questions, original image, and original question in Chinese. Please start Qwen thinking and return the thinking process:

### A.2.5 Prompts for GPT-based Scorer

This is a prompt scored with GPT4 to evaluate the scores of the two models in terms of depth and breadth.

**Prompt 6: Comparison between Different Models**

As a mathematical problem solver, please strictly compare and analyze the predictions of the two models:

[Problem description]

{question}

[Original model prediction]

{pred_raw}

[Fine-tuned model prediction]

{pred_finetuned}

Evaluation requirements:

Give the scores of raw and finetuned respectively

1. Reasoning depth score (0-3 points):

- 3 points: multi-step derivation with verification

- 2 points: complete derivation steps

- 1 point: simple calculation steps

- 0 points: no derivation process

2. Reasoning breadth score (0-3):

- 3 points: Explores multiple valid methods/angles, justifies optimal choice

- 2 points: Mentions alternative approaches briefly but focuses on one

- 1 points: Suggests another method without analysis

- 0 points: Single approach with no alternatives considered

3. Comparative analysis: use bullet points to list the main improvements/degradations

Please return a JSON that strictly follows this format: "accuracy_raw": 0-2, "accuracy_finetuned": 0-2, "depth_raw": 0-3, "depth_finetuned": 0-3, "comparison": ["Point 1", "Point 2"]

### A.2.6 Prompts for Flux Data Sythesis

We use the currently best text-to-image model **Flux** to generate images from text, and compare it with data based on ScienceQA and OKVQA as image anchors. The process involves two main steps:

**Step 1: Generating Structured Descriptions.** We first produce concise textual descriptions that serve as prompts for image generation. These descriptions are categorized into eight distinct educational domains to ensure coverage of diverse reasoning scenarios:

- **Geometry:** basic geometric shapes with labeled lengths and angles.
- **Physics:** simple mechanics such as pulleys, inclined planes, or force diagrams.
- **Chemistry:** molecular structures or reaction schemes.
- **Math Word Problem:** real-world scenes involving objects and quantities.
- **Logic Diagram:** flowcharts or condition-based logical structures.
- **Statistics Chart:** visualizations such as bar charts, pie charts, or line graphs.
- **Timeline or History Map:** event timelines or migration maps.
- **Circuit Diagram:** basic electronic circuits with labeled components.

Each category follows a unified generation prompt to ensure clarity and reliability:

**Prompt 7: Generating Image Descriptions**

For each of the following categories, generate a simple and reliable prompt for a text-to-image generation model. The goal is to create educational images with accurate content, clear structure, and moderate visual complexity.

The generated prompt should describe a plausible, textbook-style diagram with correct labels or values, suitable for multi-step reasoning (e.g. in math, science, or logic).

The content should avoid errors, and not be too crowded or complex.

Each prompt should be concise, standalone, and directly usable for image generation — do not include explanations or extra output.

> Use this category {category} and its description {description}:

**Step 2: Image Synthesis via Flux.** The resulting category-specific prompts are then input into **Flux**, which generates corresponding images with consistent style and controlled complexity. Each synthesized image is paired with its associated textual reasoning data to form complete multimodal QA samples.

This two-step pipeline—structured prompt generation followed by Flux-based synthesis—ensures both semantic precision and visual clarity, enabling the dataset to support reasoning across textual and visual modalities.

## B   Detailed Experimental Setup

### B.1   Benchmarks

To evaluate the models' reasoning capabilities and performance in both multimodal and text comprehension, we adopt the methodology used by major advanced open-source VLM Qwen2.5-VL-7B [3], selecting the following widely utilized reasoning evaluation datasets. All evaluations were conducted on the ms-swift [44] platform.

- *Multimodal Benchmarks*:
  - **MMStar** [7], an elite vision-indispensable multi-modal benchmark comprising 1,500 challenge samples meticulously selected by humans.
  - **MathVista** (Mini) [21], a benchmark designed to combine challenges from diverse mathematical and visual tasks, consists of 6,141 examples, derived from 28 existing multimodal datasets.
  - **MathVision** (Mini) [36], a meticulously curated collection of 3,040 high-quality mathematical problems with visual contexts sourced from real math competitions.

- *Text-based Benchmarks*:
  - **GSM8K** [8], a dataset of high quality linguistically diverse grade school math word problems created by human problem writers, which takes between 2 and 8 steps to solve.
  - **MATH** [12], a dataset of challenging competition mathematics problems, each problem in MATH has a full step-by-step solution.
  - **GPQA** [31], stands for Graduate-Level Google-Proof Q&A Benchmark, a challenging dataset designed to evaluate the capabilities of LLMs and scalable oversight mechanisms.

Given the extensive evaluation tasks, employing the complete evaluation sets would incur substantial time expenditure. To expedite the evaluation process while ensuring fairness and accuracy, we adopt the $eval\_limit$ parameter from ms-swift, configuring **GPQA** to 300 samples, while retaining the original sample sizes for the remaining benchmark datasets. ***All experiments were conducted using this consistent setup*** to ensure the fairness of the experiments.

### B.2   Training and Evaluation Details

**Platform**   We implement our approaches using PyTorch [28] v2.5.1, coupled with PEFT v0.14.0 and the Transformers library [39] v4.49.0. Experiments are conducted on a computing platform equipped with four NVIDIA A100 GPUs (40GB), with LLMs loaded as 16-bit floating-point numbers. The specific data-model development processes are completed in Data-Juicer Sandbox [6], via integration with the ms-swift [44] repository for training and evaluation, and the *VLMEvalKit* [9] repository for evaluation.

**Baselines**   Our method requires no external background knowledge for text generation and enhances reasoning capabilities through the model's inherent visual understanding. While this approach differs fundamentally from typical data synthesis methodologies, we select representative reasoning-enhancing datasets for comparative analysis to validate the effectiveness of our approach in SFT for VLMs.

- *Multimodal Reasoning*:
  - **MMEVOL**[23]:A method addressing data quality limitations by generating complex and diverse image-text instruction datasets to improve VLMs. We consider two subsets of **MMEVOL**: (1) For direct comparison, we utilize its **SCIENCEQA** synthetic data subset. (2) For comprehensive reasoning comparison, we adopt **DVQA** which primarily focusing on mathematical and logical reasoning tasks.
- *Text-Based Reasoning*:
  - **LIMO** [42]: Challenges conventional assumptions in mathematical reasoning by demonstrating that models achieve superior performance with smaller quantities of high-quality training data. We reformat its "question+solution+answer" structure into standard instruction pairs for training consistency.
  - **OPEN-O1** [35]: An open-source initiative to replicate the reasoning capabilities of proprietary models through curated SFT data for CoT activation. The complete 77k-sample dataset trains both LLaMA and Qwen architectures. To evaluate data efficiency, we additionally create a 400-sample randomly subsampled version for performance validation on same scale.

**Training Details**  In our experimental setup, we employ Low-Rank Adaptation (LoRA) [13] adapters for the fine-tuning process, utilizing a LoRA-rank of 8 and a LoRA-alpha of 32. The learning rate was consistently maintained at $1 \times 10^{-5}$ on Qwen2.5-VL-7B-Instruct. We utilize a batch size of 4 and set the maximum sequence length to 4096 tokens to accommodate the model's reasoning capacity. To optimize the training process, a warmup ratio of 0.05 was applied and a validation ratio of 0.03 was used. The training was conducted over a single epoch, balancing computational efficiency with the need for effective model adaptation.

In the VLM training configuration, for pure textual information, we freeze both the Vision Transformer (ViT) and the Aligner while keeping the LLM layers active, to ensure that the image processing capabilities of the VLM remain unaffected. For image-text data, we adopt the post-tuning strategy of the Qwen2.5-VL series [3], where the ViT remains frozen while both the Aligner and LLM are activated to maximize multimodal information acquisition.

**Evaluation Details**  Following the evaluation protocols established in ms-swift [44] and VLMEvalKit [9], we maintained default configurations with *pt* as the inference backend. For multimodal tasks, we implemented *VLMEvalKit* as the evaluation backend, while employing the *Native* framework for text-only evaluation scenarios.

## C   Multimodal Data Synthesis Exploration

### C.1   Methodologies

**Anchoring Image Sources.**  For the $P_{context}$ component in text-image QA synthesis, we use existing images rather than generating images from scratch (as in the text-only case). This is because the existing SOTA VLMs are mostly only image-to-text generation (i.e., image understanding) and fail to do text-to-image generation at the same time. This decision arises because current SOTA VLMs predominantly specialize in image-to-text generation (i.e., image understanding) and lack dual capabilities for text-to-image generation constrained by their architectures. Without anchored visual references, the model risks hallucinatory reasoning about image elements (e.g., referencing "growth curves" might lead to incorrect inferences about unrelated scenarios, such as spatial-based reasoning tasks).

Specifically, we randomly sample image-text pairs from scientific knowledge and general-domain QA datasets – ScienceQA [22] and OKVQA [24] – to serve as background context ($P_{context}$) for $\pi$. We emphasize alignment between visual and textual concepts within the generation instruction $P_{task}$, prompting $\pi$ to self-generate $k = 5$ seed questions per inference. This enhances harmonized interactions between image and text modalities during synthesis.

**Multimodal Cognitive Combination Types.**  For $P_{context}$ containing both text and images, we introduce four multimodal combination types:

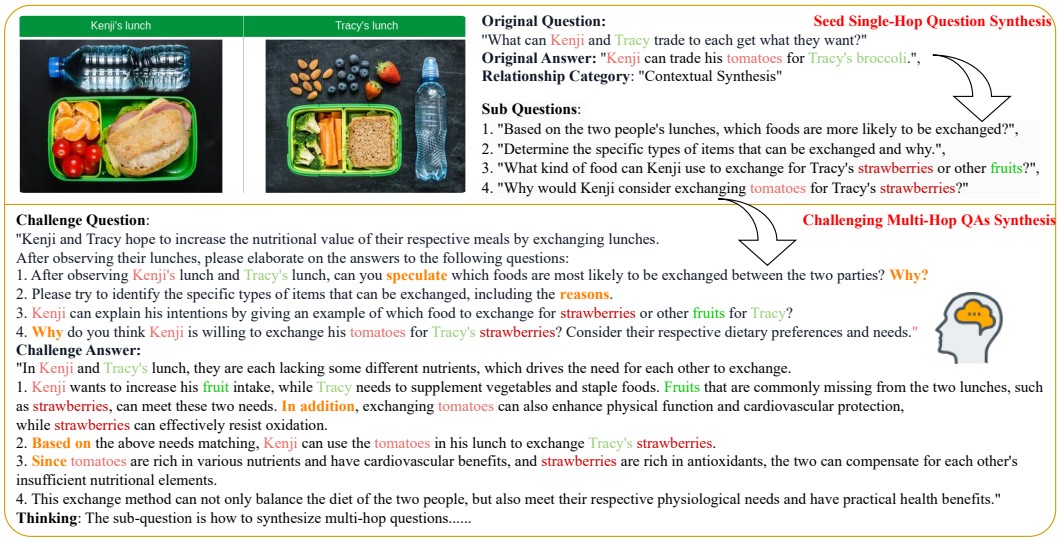

Figure 2: An end-to-end example of MINDGYM. For multimodal data, we generate five seed questions first, and make the model self-challenging itself via synthesizing multi-hop questions and multi-hop answers while preserving its internal thinking process.

- *Visual-Textual Alignment*: Ensures consistency between visual data (e.g., diagrams, charts) and textual descriptions.
- *Spatial Reasoning*: Addresses spatial relationships or geometric arrangements depicted in visual content.
- *Causal Inference*: Requires deducing cause-effect relationships from combined textual and visual inputs.
- *Contextual Synthesis*: Demands integration of information across modalities (e.g., text, images, graphs) to form unified conclusions.

Figure 2 illustrates an example in which the original $P_{\text{context}}$ (including image and textual inputs) is first used to synthesize seed questions (e.g., "What ingredients did Kenji use?" or "What vitamins are in broccoli?"), as detailed in Section 3.2). These are then combined via the multi-hop synthesis protocol (outlined in Section 3.3) into more challenging questions that require cross-contextual reasoning (e.g., "How do Kenji's lunch ingredients compare with Tracy's in terms of vitamin content?" paired with bridging and temporal relationships).

Critically, whereas the original QA focused on limited details (e.g., Kenji's tomatoes and Tracy's broccoli), our synthesized questions expand reasoning depth by incorporating broader contextual elements: sandwich compositions, citrus fruits, root vegetables, etc. The resulting multi-hop QAs demonstrate richer interconnections, broader scope, and higher cognitive demands on the solver.

## C.2 Experimental Setup

While the main body of our work focuses exclusively on synthetic textual data, we also conduct preliminary explorations into multimodal data synthesis to demonstrate the extensibility of our approach. These experiments employ the same set of vision-language models—Qwen2.5-VL-7B-Instruct, Qwen2.5-VL-32B-Instruct, InternVL-8B, and InternVL-38B—for generating synthetic image-question-answer triplets.

For the multimodal domain, we leverage widely-adopted scientific and reasoning datasets as visual source material for question synthesis, specifically including ScienceQA and OK-VQA. These datasets serve as visual grounding primitives, enabling the models to synthesize multimodal questions that align with real-world visual contexts.

- **ScienceQA** [22], the first large-scale multimodal dataset that annotates lectures and explanations for the answers.

Table 9: Performance of different models on various evaluation benchmarks. The number following the dataset name indicates its size in terms of the number of samples used for training. Different from Table 1, this table is about multimodal data synthesized by MINDGYM.

| Model Series | Dataset | Text Eval | | | | Multimodal Eval | | | | Avg |
|---|---|---|---|---|---|---|---|---|---|---|
| | | GSM8K | Math | GPQA | Text-Avg | MMStar | MathVista | MathVision | MM-Avg | |
| Qwen2.5-VL-7B | raw | 83.62 | 67.60 | 31.83 | 61.02 | 64.00 | 69.30 | 24.67 | 52.66 | 56.84 |
| | Openo1-sft (400) | 84.31 | 69.00 | 29.70 | 61.00 | 63.87 | 69.70 | 23.66 | 53.08 | 57.04 |
| | Openo1-sft (4k) | 77.94 | 59.00 | 28.19 | 55.04 | 58.93 | 61.40 | 17.43 | 45.92 | 50.48 |
| | LIMO (817) | 84.08 | 67.80 | 30.58 | 60.82 | 63.93 | 70.20 | 25.90 | 53.37 | 57.10 |
| | MMEvol-SciQA (106) | 83.40 | 65.60 | 31.83 | 60.28 | 63.93 | 69.50 | 23.36 | 52.26 | 56.27 |
| | MMEvol-DvQA (4k) | 83.85 | 65.80 | 33.08 | 60.91 | 62.93 | 67.40 | 24.01 | 51.45 | 56.18 |
| | **MindGYM-OKVQA(ours)** | **84.00** | 68.2 | 31.08 | 61.09 | 63.60 | 69.20 | **28.29** | 53.70 | **57.40** |
| | **MindGYM-SciQA(ours)** | 83.93 | **69.00** | 31.45 | **61.46** | 64.20 | 69.80 | 25.33 | 53.11 | 57.29 |
| Qwen2.5-VL-32B | raw | 95.15 | 81.80 | 47.98 | 74.98 | 69.60 | 73.40 | 37.50 | 60.17 | 67.58 |
| | Openo1-sft (400) | 95.38 | 81.20 | 50.51 | 75.70 | 68.80 | 73.40 | 36.18 | 59.46 | 67.58 |
| | Openo1-sft (4k) | 95.68 | 79.80 | 38.00 | 71.16 | 68.27 | 72.20 | 34.54 | 58.34 | 64.75 |
| | LIMO (817) | 95.83 | 80.80 | 41.92 | 72.85 | 69.33 | 71.80 | 37.83 | 59.65 | 66.25 |
| | MMEvol-SciQA (106) | 95.60 | 81.00 | 42.42 | 73.01 | 69.00 | 73.80 | 36.84 | 59.88 | 66.44 |
| | MMEvol-DvQA (4k) | 92.72 | 80.60 | 38.38 | 70.57 | 69.20 | 72.60 | **39.14** | 60.31 | 65.44 |
| | **MindGYM-OKVQA(ours)** | 95.53 | **82.20** | 47.47 | 75.07 | **69.60** | 73.20 | **39.14** | **60.65** | **67.86** |
| | **MindGYM-SciQA(ours)** | 95.45 | 81.20 | 47.98 | 74.88 | 68.87 | 73.20 | 38.82 | 60.30 | 67.59 |
| InternVL-8B | raw | 86.50 | 76.30 | 33.84 | 66.45 | 69.00 | 73.20 | 33.22 | 58.47 | 62.46 |
| | Openo1-sft (400) | 89.39 | 77.20 | 43.43 | 70.01 | 69.00 | 73.60 | 32.57 | 58.39 | 64.20 |
| | Openo1-sft (4000) | 88.55 | 78.00 | 42.42 | 69.66 | 68.87 | 72.60 | 31.91 | 57.79 | 63.72 |
| | LIMO (817) | 89.16 | 76.40 | 42.42 | 69.33 | 69.00 | 73.40 | 32.57 | 58.32 | 63.82 |
| | MMEvol-SciQA (106) | 88.40 | 77.80 | 46.97 | 71.06 | 69.13 | 72.80 | 33.88 | 58.60 | 64.83 |
| | MMEvol-DvQA (4k) | 88.55 | 74.60 | 40.40 | 67.85 | 68.47 | 71.60 | 32.24 | 57.44 | 62.65 |
| | **MindGYM-OKVQA(ours)** | 88.48 | **78.40** | 43.94 | 70.27 | 69.07 | 73.10 | 31.25 | 57.81 | 64.04 |
| | **MindGYM-SciQA(ours)** | 89.16 | 77.60 | 42.42 | 68.45 | 69.27 | 74.20 | 32.89 | 58.79 | 63.62 |
| InternVL-38B | raw | 89.16 | 72.60 | 47.47 | 69.74 | 72.40 | 72.80 | 36.51 | 60.57 | 65.16 |
| | Openo1-sft (400) | 89.46 | 76.4 | 48.99 | **71.62** | 72.27 | 72.00 | 35.53 | 59.93 | 65.77 |
| | Openo1-sft (4k) | 89.46 | 79.20 | 44.95 | 71.20 | 72.47 | 72.90 | 35.53 | 60.30 | 65.75 |
| | LIMO (817) | 89.16 | 72.60 | 47.47 | 69.74 | 72.40 | 72.80 | 36.51 | 60.57 | 65.16 |
| | MMEvol-SciQA (106) | 89.46 | 75.00 | 46.46 | 70.31 | 72.40 | 73.20 | 38.49 | 61.36 | 65.83 |
| | MMEvol-DvQA (4k)} | 89.92 | 74.00 | 46.46 | 70.13 | 70.93 | 71.70 | 30.92 | 57.85 | 63.99 |
| | **MindGYM-OKVQA(ours)** | 88.93 | 75.60 | **48.99** | 71.17 | **72.53** | 71.80 | 36.18 | 60.17 | 65.67 |
| | **MindGYM-SciQA(ours)** | 89.31 | 75.60 | 48.48 | 71.13 | 72.33 | 72.60 | 35.86 | 60.26 | 65.70 |

- **OK-VQA** [24], a dataset for visual question answering that requires methods which can draw upon outside knowledge to answer questions.

During training with image-text data, we update only the Aligner-layers, keeping both the ViT and LLM layers frozen to preserve pretrained representations and ensure stable adaptation. These results are not the main focus of the current study but demonstrate the extensibility of our method to multimodal contexts.

### C.3 Experimental Results

**Cross-modal Transfer.** MINDGYM enables effective cross-modal reasoning capability sharing through bidirectional knowledge transfer. Remarkably, the text-specialized MINDGYM-Text establishes new SOTA performance on MM-AVG, with the score of **54.42** that surpasses all baselines. Conversely, the multimodal-enhanced MINDGYM-SciQA demonstrates unexpected supremacy in textual reasoning tasks, achieving an optimal TEXT-AVG score of **61.46**. This bidirectional enhancement confirms the transferability of reasoning capabilities between textual and multimodal representations.

**Generalizability & Robustness.** Consistent improvements emerge across diverse data sources. Our approach attains sub-optimal average performance when synthesizing data from both SciQA, showing an improvement of **0.45** over raw data, and OKVQA, with an improvement of **0.56** over raw data, demonstrating adaptability to various domain-specific reasoning tasks. This indicates that our method effectively enhances generalizability and robustness across diverse reasoning challenges.

### C.4 Data Analysis

We conduct a comparative analysis across baselines and our MindGYM-generated datasets to assess the impact of cognitively guided data on reasoning quality and complexity. As shown in Table 10,

Table 10: Data-Juicer analysis results across different models. Different from Table 2, this table is about multimodal data synthesized by MINDGYM.

| Model | Dataset | quality-mean | quality-std | action | dependency | token | length |
|---|---|---|---|---|---|---|---|
| baseline | Openo1-sft (400) | 0.96 | 0.091 | 8.05 | 2.00 | 77 | 274 |
| | Openo1-sft (4k) | 0.70 | 0.10 | 8.78 | 2.02 | 83 | 284 |
| | LIMO (817) | 0.87 | 0.10 | 7.71 | 2.02 | 101 | 322 |
| | MMEvol-SciQA (106) | 0.91 | 0.082 | 2.31 | 1.85 | 13.73 | 67 |
| | MMEvol-DvQA (4k) | 0.76 | 0.11 | 2.66 | 1.95 | 14.82 | 69 |
| Qwen2.5-VL-7B | MindGYM-OKVQA | 0.93 | 0.079 | 9.04 | 2.30 | 122 | 96 |
| | MindGYM-SciQA | 0.90 | 0.099 | 10.23 | 2.27 | 144 | 117 |
| Qwen2.5-VL-32B | MindGYM-OKVQA | 0.97 | 0.044 | 10.94 | 2.25 | 168 | 134 |
| | MindGYM-SciQA | 0.98 | 0.031 | 15.44 | 2.16 | 233 | 195 |
| InternVL-8B | MindGYM-OKVQA | 0.96 | 0.058 | 9.62 | 2.32 | 141 | 110 |
| | MindGYM-SciQA | 0.96 | 0.045 | 10.64 | 2.26 | 148 | 118 |
| InternVL-38B | MindGYM-OKVQA | 0.74 | 0.069 | 7.19 | 2.28 | 111 | 91 |
| | MindGYM-SciQA | 0.97 | 0.039 | 9.26 | 2.23 | 131 | 119 |

| | Datasets | Multimodal Eval | | | | Text-Only Eval | | | | AVG |
|---|---|---|---|---|---|---|---|---|---|---|
| | | MMStar | MathVista | MathVision | MM-AVG | GSM8K | MATH | GPQA | TEXT-AVG | |
| Baslines | Raw | 64.00 | 69.30 | 24.67 | 52.66 | 83.62 | 67.60 | 31.83 | 61.02 | 56.84 |
| | LIMO (817) [42] | 63.93 | 70.20 | 25.90 | 53.37 | 84.08 | 67.80 | 30.58 | 60.82 | 57.10 |
| | OPEN-O1 (77K) [35] | 58.93 | 61.40 | 17.43 | 45.92 | 77.94 | 59.00 | 28.19 | 55.04 | 50.48 |
| | OPEN-O1 (400) [35] | 63.87 | 69.70 | 23.66 | 53.08 | 84.31 | 69.00 | 29.70 | 61.00 | 57.04 |
| | MMEVOL-SCIQA (106) [23] | 63.93 | 69.50 | 23.36 | 52.26 | 83.40 | 65.60 | 60.28 | 60.28 | 56.27 |
| | MMEVOL-DVQA (4K) [23] | 62.93 | 67.40 | 24.01 | 51.45 | 83.85 | 65.80 | 33.08 | 60.91 | 56.18 |
| Text | MINDGYM | 64.33 | 70.30 | 28.62 | 54.42 | 84.08 | 68.40 | 31.33 | 61.27 | 57.84 |
| | MINDGYM w/o SC | 63.60 | 69.90 | 24.67 | 52.72 | 83.83 | 66.60 | 29.57 | 60.00 | 56.36 |
| | MINDGYM Syn-EN | 63.93 | 69.70 | 24.67 | 52.77 | 84.00 | 67.20 | 29.70 | 60.30 | 56.53 |
| | MINDGYM w/o OF | 64.00 | 69.90 | 22.69 | 52.20 | 84.15 | 67.00 | 31.33 | 60.83 | 56.51 |
| | MINDGYM w RB | 64.00 | 70.10 | 22.70 | 52.20 | 84.15 | 67.00 | 31.33 | 60.83 | 56.51 |
| Image | MINDGYM-OKVQA | 63.60 | 69.20 | 28.29 | 53.70 | 84.00 | 68.20 | 31.08 | 61.09 | 57.40 |
| | MINDGYM-OKVQA w/o SC | 63.93 | 69.30 | 25.33 | 52.85 | 84.15 | 68.80 | 30.95 | 61.30 | 57.08 |
| | MINDGYM-OKVQA Syn-EN | 63.53 | 69.90 | 25.99 | 53.14 | 84.23 | 67.60 | 29.95 | 60.59 | 56.87 |
| | MINDGYM-OKVQA w RB | 63.80 | 71.20 | 25.00 | 53.33 | 84.00 | 66.60 | 31.20 | 60.60 | 56.97 |
| | MINDGYM-SciQA | 64.20 | 69.80 | 25.33 | 53.11 | 83.93 | 69.00 | 31.45 | 61.46 | 57.29 |
| | MINDGYM-SciQA w/o SC | 64.07 | 69.20 | 26.32 | 53.20 | 83.93 | 67.00 | 30.95 | 60.63 | 56.91 |
| | MINDGYM-SciQA Syn-EN | 63.73 | 69.00 | 24.01 | 52.25 | 83.55 | 68.00 | 31.08 | 60.88 | 56.56 |
| | MINDGYM-SciQA w RB | 64.20 | 70.20 | 25.99 | 53.46 | 83.85 | 65.00 | 31.96 | 60.27 | 56.87 |

Table 11: Complete ablation study results of removing *stream of consciousness* (*w/o SC*), utilizing English in data synthesis (*Syn-EN*), changing the order of fine-tuning (*w/o OF*) steps and Relation Balanced (*with RB*).

MindGYM datasets (OKVQA and SciQA) consistently yield higher reasoning quality scores, with Qwen2.5-VL-32B achieving the best performance (0.98 quality mean with 0.031 std on MindGYM-SciQA), significantly outperforming standard instruction-tuning sets such as Openo1-sft (0.70) and LIMO (0.87). In addition, MindGYM prompts models to generate substantially longer and more structured reasoning traces, as evidenced by the increase in action steps (up to 15.44) and dependency depth (avg. >2.2). This indicates that our cognitively annotated data effectively stimulates deeper reasoning behaviors without sacrificing generation stability. Notably, while token and input length grow with model capacity, the reasoning quality remains robust, suggesting that MindGYM scales well with model size and supports more abstract reasoning. In contrast, traditional instruction datasets lead to flatter, shorter outputs with limited cognitive structure. These results demonstrate the value of our data-centric approach in enhancing model reasoning ability and interpretability.

## D  Comprehensive Ablation Studies

### D.1  Complete Experimental Results

We present the complete version of Table 4 in Table 11, which demonstrates the comprehensive ablation study results of our method. In addition to conducting ablation experiments on the text-based

variant of our approach (as described in Appendix C), we also performed an ablation study on the multimodal data synthesis methodology. As shown in Table 11, the performance degradation observed in the OKVQA and SciQA ablation experiments compared to the full implementation of our method provides compelling evidence for two key aspects: 1) the effectiveness of our proposed framework, and 2) its cross-modal generalization capability across different data modalities. This empirical validation underscores the importance of our multimodal synthesis strategy in maintaining model performance across diverse vision-language tasks.

## D.2 Comparison with Recent Proprietary Systems

To situate MindGym's data-driven improvements in context, we compare the fine-tuned **Qwen2.5-VL-72B-MindGym** model with leading proprietary reasoning systems on the **MathVision** leaderboard. While MindGym is not a new architecture but a synthesis framework, it delivers a consistent +1.37% gain over the original Qwen2.5-VL model, narrowing the gap to top commercial models. Closed-source systems such as **o3** and **o4** remain inaccessible for fine-tuning, but MindGym effectively enhances open-source models under comparable evaluation settings.

Table 12: MathVision leaderboard comparison.

| Model | Score |
|---|---|
| GPT-4.5 | 47.3 |
| Gemini-2 Flash | 41.3 |
| **Qwen2.5-VL-72B-MindGym** | **39.47** |
| Kimi k1.5 | 38.6 |
| **Qwen2.5-VL-72B (baseline)** | **38.10** |
| Claude 3.5 Sonnet | 37.99 |
| Kimi-VL-A3B-Thinking | 36.8 |

## D.3 Comparison with MathFusion

MindGYM is conceptually related to MathFusion [29] in that both approaches leverage synthetic data to improve reasoning capabilities. However, MindGYM extends this idea by integrating **multimodal and cognitively structured synthesis**, covering a wider range of domains beyond purely mathematical tasks. In particular, the *Thinking Claude* framework embeds structured reasoning trajectories into both questions and answers, providing richer supervision for reasoning-aware fine-tuning.

To illustrate the difference, we compared a MathFusion-style synthesis with MindGYM on the same evaluation benchmarks:

Table 13: Comparison between MathFusion-style synthesis and MindGYM.

| Method | GSM8K | Math | GPQA | text-avg | MMStar | MathVista | MathVision | MM-avg | Overall Avg |
|---|---|---|---|---|---|---|---|---|---|
| MathFusion-style | 84.31 | 69.1 | 29.0 | 60.80 | 61.43 | 67.2 | 24.97 | 51.20 | 56.00 |
| MindGYM | 84.08 | 68.4 | 31.33 | 61.27 | 64.33 | 70.3 | 28.62 | 54.42 | **57.84** |

As shown in Table 13, MindGYM outperforms the MathFusion-style baseline on multimodal and general reasoning tasks, demonstrating the effectiveness of combining cognitive structure with multimodal synthesis.

