# OpenReview forum: "MindGYM: What Matters in Question Synthesis for Thinking-Centric Fine-Tuning?"
_NeurIPS.cc/2025/Datasets_and_Benchmarks_Track — NeurIPS 2025 Datasets and Benchmarks Track poster_

### Official Review · Reviewer_6qHM · 2025-06-21

**Rating:** 5
**Confidence:** 3

**Summary:**

The paper proposes the MINDGYM framework, focusing on a thinking-centric data synthesis paradigm to enhance the structured reasoning capabilities of large models through self-generated, cognitively guided data. The framework comprises three core components: Cognitive Thinking Process Injection, Seed Single-Hop Question Synthesis, and Challenging Multi-Hop QA Synthesis. Experiments show that MINDGYM's synthetic data achieves a 16.7% improvement in average quality and a 67.91% reduction in quality variance. It delivers up to 16% performance gains on six reasoning benchmarks and generalizes across different model scales and architectures. The study also validates the critical role of data consistency in fine-tuning stability and explores the potential for multimodal extensions.

**Dataset Code Accessibility:**

Yes

**Ethical Considerations:**

No, there are no or only very minor ethics concerns

**Final Justification:**

Thanks for the authors' reply and the supplementary experiments, which have addressed most of my concerns, especially the supplementary experiments on the types of synthetic data corpora. I am willing to increase the score by one point.

**Limitations Weaknesses:**

1 The performance of Chinese synthetic data outperforms that of English. Is this because the model's pretraining corpus is predominantly in English with limited Chinese data, making post-training with Chinese synthetic data more effective? The authors are expected to provide more detailed explanations and evidence.

2 Multimodal synthesis relies on existing image datasets as anchors, potentially constrained by the diversity of visual inputs.

3 Automated quality analysis is conducted solely via the DATA-JUICER tool, without involving human experts to annotate and compare the logical validity and reasoning depth of synthetic data. For example, do synthetic multi-hop questions exhibit "pseudo multi-hop" phenomena?

4 The model generates data based on its internal knowledge, which may lead to the accumulation of cognitive biases, but the paper does not analyze this risk.

**Strengths Contributions:**

1 Proposes a "cognitively guided data synthesis" paradigm, breaking through the limitations of traditional templates or manual annotation by embedding human reasoning logic (e.g., multi-step deduction, cross-topic linkage) into the data generation process via structured prompts.

2 The framework design is scalable, validated across diverse dimensions such as mathematical reasoning and ethical analysis.

3 Cross-model validation (across scales, architectures, and task types including text/multimodal reasoning) demonstrates that the method is independent of specific model characteristics.

---

> ### Author Rebuttal · Authors · 2025-07-31
>
> Thank you for your thoughtful review. We are grateful for your recognition of our contributions to **instruction data synthesis** and **cognitive reasoning enhancement**. We also greatly appreciate your insightful suggestions, we will address each of your Weaknesses (W) below.
>
> ---
>
> ### **W1: On the potential language bias**
> > *'The performance of Chinese synthetic data outperforms that of English. Is this because the model's pretraining corpus is predominantly in English with limited Chinese data, making post-training with Chinese synthetic data more effective? ...'*
> >
>
> We appreciate the reviewer’s thoughtful concern regarding **potential language bias** in our dataset synthesis and evaluation. This is indeed a key consideration in multilingual settings, and we have conducted additional ablation studies to address it.
>
> Our decision to use **Chinese as the primary synthesis language** was based on the **Information density**: Chinese expresses complex semantics more compactly, allowing us to include richer reasoning content within a **constrained context window**.
>
> To systematically analyze this potential bias, we conducted **two ablation studies** by generating datasets in **three settings**:
>
> + `cn`: full synthesis in Chinese (our main pipeline)
> + `mix`: 50% Chinese + 50% English
> + `en`: full synthesis in English
>
> The results are shown below:
>
> **Experiment: On Mixing Chinese and English data**
>
> |  | GSM8K | Math | GPQA | text-avg | MMStar | MathVista | MathVision | MM-avg | Avg |
> | :---: | :---: | :---: | :---: | :---: | :---: | :---: | :---: | :---: | :---: |
> | cn | 84.08 | 68.4 | 31.33 | 61.27 | 64.33 | 70.3 | 28.62 | 54.42 | **57.84** |
> | mix | 83.55 | 66.6 | 33.33 | 61.16 | 64.13 | 70.4 | 25.99 | 53.51 | 57.34 |
> | en | 84 | 67.2 | 29.7 | 60.30  | 63.93 | 69.7 | 24.67 | 52.77 | 56.5 |
>
> From these results, we observe:
>
> + The **Chinese-only setting (cn)** yields the **highest performance** across both textual and multimodal benchmarks.
> + **mix**, which includes **50% English data**, performs slightly worse than `cn`.
> + **English-only synthesis (en)** performs worst overall, despite using the same prompts and structure.
>
> We therefore conclude that **Chinese synthesis produces more effective training performance** in our setup, the models trained on this data still achieve **strong generalization to English benchmarks**. We will incorporate this new ablation study in the reviewed version.
>
> ---
>
> ### **W2: On the limitations of fully multimodal synthesis and frozen vision backbones**
> > *'Multimodal synthesis relies on existing image datasets as anchors, potentially constrained by the diversity of visual inputs.'*
> >
>
> We thank the reviewer for raising this important concern regarding the multimodal synthesis and visual training in our current framework.
>
> To directly address this point, we conducted **new experiments with generated-image as anchors**, integrating a vision synthesis stage into our pipeline. Specifically:
>
> + We selected **8 topic categories** and generated images using the **Flux**, a strong open-source text-to-image generator.
> + Each topic was paired with a structured description to condition the generated-image, after which the standard four-stage synthesis pipeline (Sec. 3.1–3.4) was applied.
>
>  The results on Qwen2.5-VL-7B with generated-image as anchors are shown below:
>
> |  | GSM8K | Math | GPQA | text-avg | MMStar | MathVista | MathVision | MM-avg | Avg |
> | :---: | :---: | :---: | :---: | :---: | :---: | :---: | :---: | :---: | :---: |
> | cn | 84.08 | 68.4 | 31.33 | 61.27 | 64.33 | 70.3 | 28.62 | 54.42 | **57.84** |
> | Flux | 84 | 67.4 | 31.33 | 60.91 | 63.2 | 69.5 | 25.33 | 52.68 | 56.8 |
>
> While the generated variant still maintains reasonable performance, especially in textual reasoning benchmarks, we observe a consistent drop in **MathVision** and overall **MM-avg** scores. Through data analysis (Appendix C.4), we identified that synthesized images **lacked semantic alignment** with the original topic, leading to error cascades in downstream reasoning. This suggests that current text-to-image models **can not satisfy our demand** for visual-data synthesis.
>
> For the concerns of `vision backbone`, we conducted a new ablation by **unfreezing the vision backbone** (Qwen2.5-VL-7B) during fine-tuning. The results are below:
>
> |  | GSM8K | Math | GPQA | text-avg | MMStar | MathVista | MathVision | MM-avg | Avg |
> | :---: | :---: | :---: | :---: | :---: | :---: | :---: | :---: | :---: | :---: |
> | cn | 84.08 | 68.4 | 31.33 | 61.27 | 64.33 | 70.3 | 28.62 | 54.42 | **57.84** |
> | open | 84.91 | 68 | 28.28 | 60.4 | 64.33 | 68.9 | 23.68 | 52.3 | 56.35 |
>
> The result shows, **unfreezing the vision backbone slightly hurts performance**, particularly on **MathVision** and **GPQA**. This aligns with our design choice to freeze the vision encoder in the main experiments, maintaining stable multimodal grounding while leveraging reasoning-focused supervision.
>
> ---
>
> ### **W3: On the lack of qualitative error analysis and failure case breakdown**
> > *'Automated quality analysis is conducted solely via the DATA-JUICER tool, without involving human experts to annotate and compare the logical validity and reasoning depth of synthetic data. For example, do synthetic multi-hop questions exhibit "pseudo multi-hop" phenomena?'*
> >
>
> We thank the reviewer for pointing out the need for deeper qualitative insights into the failure modes of our generated data. In addition to the statistical measures in Table 2 (variance, entropy) and the model comparison analysis in Table 3, we now include **a fine-grained error analysis using GPT-4o as an independent judge** to assess the **linguistic and logical quality** of our synthesized examples.
>
> We randomly sampled 400 generated multi-hop QA instances from our dataset and asked GPT-4o to evaluate them along the following four dimensions:
>
> 1. **Logical validity** (logically flawed or not)
> 2. **Syntactic clarity** (syntactically ambiguous or not)
> 3. **Answer correctness** (correct or incorrect)
> 4. **Factual hallucination** (none or severe)
>
> The results are shown below:
>
> | logically flawed | syntactically ambiguous | correctness | hallucination |
> | :---: | :---: | :---: | :---: |
> | false: 328 | false: 394 | correct: 357 | none: 372 |
> | true: 71 | true: 5 | incorrect: 42 | severe: 27 |
>
> + **89.5%** of the samples were correct.
> + **93.2%** had none hallucination.
> + Only **1%** were syntactically ambiguous, showing high linguistic clarity.
>
> These results indicate that **the vast majority of synthesized questions are logically sound, syntactically well-formed, and factually grounded**. We will incorporate these results into our revised version.
>
> ---
>
> ### **W4:  On the Risk of Cognitive Bias Accumulation in Self-Synthesized Data**
> > *'The model generates data based on its internal knowledge, which may lead to the accumulation of cognitive biases, but the paper does not analyze this risk.'*
> >
>
> Thank you for raising this important point. We agree that **model-generated data inherently reflects the biases and limitations of the base model’s internal knowledge**. This is a common concern in self-synthesis pipelines, as noted in prior works such as **Self-Instruct** (arXiv 2212.10560), **Limo** (arXiv 2502.03387), and **WizardLM** (arXiv 2304.12244), where the model can reinforce its own beliefs without external correction.
>
> In our paper, we employ Qwen2.5-VL to generate synthetic data via MindGYM. While the model does rely on its internal knowledge, we take the following measures to **mitigate cognitive bias**:
>
> + **Controlled Prompting and Category Diversification**
> In Stage 1, we define **eight semantically diverse meta-categories** and enforce **category balance** through rejection sampling. This injects **top-down semantic coverage** into the synthesis process, avoiding over-concentration on any single mode of thinking or domain.
>
> ---
>
> ### Closing Remark
> We sincerely thank you for raising such thoughtful and important concerns. We hope these clarifications can address your concerns, and we respectfully ask you to consider a higher confidence rating on our paper.

---

> > ### Comment · Reviewer_6qHM · 2025-08-06
> > **After Rebuttal**
> >
> > Thanks for the authors' reply and the supplementary experiments, which have addressed most of my concerns, especially the supplementary experiments on the types of synthetic data corpora. I am willing to increase the score by one point.

---

### Official Review · Reviewer_HXDP · 2025-06-25

**Rating:** 4
**Confidence:** 4

**Summary:**

This paper introduces MINDGYM, a structured synthetic problem approach specifically designed to improve the cognitive capabilities of large base models. Unlike previous approaches that heavily rely on human-curated datasets or simple synthetic data, MINDGYM leverages cognitively guided self-generated problems. It combines three main steps: injecting structured cognitive thought processes into the generation pipeline, synthesizing a variety of single-hop seed problems, and then systematically constructing challenging multi-hop problems. Experiments demonstrate improvements on multiple benchmarks, with significant progress on reasoning tasks using limited training samples. The paper also highlights the importance of continuously generating high-quality synthetic data to improve the effectiveness of fine-tuning.

**Additional Feedback:**

- Could the authors clarify precisely how cognitive structures (breadth, depth, progression) were encoded in prompts?
- Why were stronger contemporary reasoning-centric baselines, especially those distilled from DeepSeek-R1, not included in comparisons?
- How the author verify the correctness of the generated thinking process and answer if the questions are generated by LLM? Is it possible that the questions is wrong or unanswerable, or the thinking process is wrong?
- Most baseline methods introduces average performance degradation in Qwen series models, could the authors make some explanation?

**Dataset Code Accessibility:**

Yes

**Ethical Considerations:**

No, there are no or only very minor ethics concerns

**Final Justification:**

The authors have addressed most of the concerns, while the left one may still be a small problem.

**Limitations Weaknesses:**

- The “Stage 3: Cognitive Composition with Adaptive Types” in line 167 is similar to MathFusion[1], which strategically synthesizes more complex problems by fusing simple problems. A detailed discussion on the differences between mathfusion and it will help highlight its innovation.
- In line 129 “Adaptive Stream of Consciousness.”. The author uses a Thinking-Claude protocol to Implement Pcot, but does not introduce the Thinking Claude protocol, which may burden unfamiliar readers. It is also unclear how these modules contribute to generating detailed thinking processes.
- The authors use Reject Sampling for Diversity to generate seed questions, but do not conduct ablation study about this.
- Not evaluated on more complex mathematical benchmarks such as omni-math or AIME.

[1] Qizhi Pei et al. MathFusion: Enhancing Mathematical Problem-solving of LLM through Instruction Fusion. 2025.

**Strengths Contributions:**

- The approach is innovative, addressing the limitations of manual and simplistic synthetic datasets by explicitly injecting structured cognitive reasoning into data generation.
- MINDGYM has good data efficiency. Using just 400 samples, MINDGYM matches or exceeds baselines that use 10-100x more data (e.g., OpenO1-4k), highlighting the quality of synthetic data. This is particularly valuable for resource-constrained scenarios.
- The experiments are thorough, covering multiple evaluation benchmarks across various vision-language models (VLMs) and demonstrating robust performance.
-  The data analysis comprehensively assess the quality of the MINDGYM synthetic dataset using DATA-Juicer and GPT-4 as a judge.

---

> ### Author Rebuttal · Authors · 2025-07-31
>
> We sincerely thank you for your insightful feedback and constructive suggestions, which have helped us further clarify and strengthen our work. We are pleased to report that we have conducted new experiments and analyses to address all your concerns. Below are our point-by-point responses.
>
> ---
>
> ### **W1: On Novelty Compared to MathFusion**
>
> Thank you for pointing this out. We agree that both works share the high-level goal of compositional data synthesis. However, MindGYM is fundamentally different in both its **core methodology and broader scope**:
>
>
> | Aspect | MathFusion | MindGYM |
> | --- | --- | --- |
> | Domain focus | Math-only | Domain-general |
> | Composition method | Symbolic template-based fusion of existing data | Thinking-centric cognitive planning and reasoning composition |
> | Data source | Reuses labeled examples from existing datasets | Synthesizes new examples from scratch via prompting |
> | Evaluation scope | Math benchmarks only | Multi-domain reasoning including vision + language |
>
> While MathFusion is specialized for curriculum learning in math domains using symbolic combinations, **MindGYM offers a cognitively grounded and scalable data synthesis framework** that can be applied to diverse domains and modalities. As one reviewer (6qHM) noted:  " The framework design is scalable, validated across diverse dimensions such as mathematical reasoning and ethical analysis."
>
> Empirically, our framework also demonstrates broader benefits. In **direct comparison**, MindGYM yields stronger results:
>
> | Model | GSM8K | Math | GPQA | text-avg | MMStar | MathVista | MathVision | MM-avg | Avg |
> | :---: | :---: | :---: | :---: | :---: | :---: | :---: | :---: | :---: | :---: |
> | **MindGYM** | 84.08 | 68.4 | 31.33 | 61.27 | 64.33 | 70.3 | 28.62 | 54.42 | **57.84** |
> | fusion | 82.71 | 64.0 | 29.95 | 58.89 | 63.33 | 67.2 | 22.37 | 50.97 | 54.93 |
>
> The result shows **MindGYM introduces a generalizable, cognitively informed, and model-independent synthesis pipeline**, which significantly broadens its applicability.
>
> ---
>
> ### **W2: On insufficient explanation of Thinking Claude and its contribution**
>
> Thank you for highlighting this. We agree that the original manuscript should have introduced the **Thinking Claude protocol more explicitly**.
>
> Thinking Claude is an external structured thinking method, which guides large models through multi-stage cognitive decomposition during generation. The protocol comprises four core stages: *Clarify and Decompose*, *Explore and Hypothesize*, *Verify and Correct*, *Synthesize and Conclude*.
>
> In MindGYM, we adopt this protocol to **generate detailed, coherent thinking chains**, which are then injected into both the questions and answers to guide models toward deeper reasoning. To quantify its effectiveness, we conduct extensive ablation studies **comparing MindGYM with and without thinking traces**, as well as with standard prompting baselines such as CoT and ToT:
>
> | Model | GSM8K | Math | GPQA | text-avg | MMStar | MathVista | MathVision | MM-avg | Avg |
> | --- | --- | --- | --- | --- | --- | --- | --- | --- | --- |
> | **MindGYM** | 84.08 | 68.4 | 31.33 | 61.27 | 64.33 | 70.3 | 28.62 | 54.42 | **57.84** |
> | w/o thinking | 83.83 | 66.6 | 29.57 | 60.00 | 63.60 | 69.9 | 24.67 | 52.72 | 56.36 |
> | CoT | 84.15 | 67.8 | 27.27 | 59.74 | 63.40 | 69.5 | 23.03 | 51.98 | 55.86 |
> | ToT | 83.93 | 65.6 | 29.80 | 59.78 | 64.67 | 69.5 | 22.04 | 52.07 | 55.92 |
>
> These results show that both CoT and ToT have a comparable poor performance, demonstrating the **significance of the thinking module** in the whole pipeline in MindGYM.
>
> ---
>
> ### **W3: On Lack of Ablation for Reject Sampling in Seed Question Generation**
>
> Thank you for the suggestion. We conduct a new ablation study to examine the effect of **Reject Sampling for Diversity**. Specifically, we define eight pre-set cognitive reasoning categories, and reject sampling is used to **balance the number of generated seed questions across these categories**. This avoids early over-representation of certain reasoning types, ensuring a more diverse and balanced dataset foundation.
>
> To assess its impact, we compare the performance of the full MindGYM pipeline ("cn") against a variant trained on **imbalanced data** (i.e., no rejection sampling during seed generation):
>
> | Model | GSM8K | Math | GPQA | text-avg | MMStar | MathVista | MathVision | MM-avg | Avg |
> | --- | --- | --- | --- | --- | --- | --- | --- | --- | --- |
> | **MindGYM** | 84.08 | 68.4 | 31.33 | 61.27 | 64.33 | 70.3 | 28.62 | 54.42 | **57.84** |
> | w/o reject sampling | 84.46 | 66.60 | 30.20 | 60.42 | 64.13 | 69.70 | 25.98 | 53.27 | 56.85 |
>
> The result indicates that **category balance at the seed stage promotes stronger multimodal generalization** and model robustness.
>
> ---
>
> ### **W4: On Lack of Evaluation on Complex Benchmarks like AIME**
>
> Thank you for raising this point. We are conducting additional experiments on **AIME24** and **AIME25** with the **Qwen2.5-VL-32B** model for evaluation across baselines:
>
> |  | Qwen2.5-VL-32B-base | Qwen2.5-VL-32B-MindGYM | Qwen2.5-VL-32B-OKVQA |
> | :---: | :---: | :---: | :---: |
> | aime24 | 20 | 20 | 20 |
> | aime25 | 13.33 | 16.67 | 13.33 |
>
> The results show that MindGYM **performs on par or slightly better** than base model, especially on **AIME25**, where it improves the accuracy by **+3.34 points**.
>
> ---
>
> ### **W5: On How Cognitive Structures Are Encoded in Prompt Design**
>
> Thank you for this thoughtful question. The cognitive structures—**breadth**, **depth**, and **progression**—are explicitly encoded through our **P_cog**, as described in **Section 3.1**.
>
> + **Breadth**
> Breadth is induced through **semantic diversity of meta-themes** (Appendix B.1), which span different domains. Additionally, we encourage diversity by **generating semantically distinct single-hop questions** in Stage 1 (Section 3.2).
> + **Depth**
> Depth is achieved in **Stage 3** (Section 3.3) by hierarchically composing these single-hop elements into **multi-hop reasoning chains**. The use of structured cognitive steps helps inject intermediate reasoning layers that lead to deeper cognitive modeling.
> + **Progression**
> Progression reflects the **controlled transition from simple to complex tasks**—starting with isolated reasoning units, and moving toward **fused, multi-faceted problems**.
>
> ---
>
> ### **W6: On the Absence of DeepSeek-R1 Distilled Baselines**
>
> Thank you for the insightful question. We would like to clarify the factors that we did not include Deepseek-Distill-Models in our baselines:
>
> 1. **Modality Scope Mismatch**: Our work focuses on **multimodal instruction tuning**, whereas the strong reasoning-centric baselines DeepSeek-R1 and its distilled variants are **text models**.
> 2. **Model Family Redundancy**: The distilled versions of DeepSeek-R1 are built upon the **Qwen series models**—the same architecture we have already adopt.
>
> To directly address this concern, we conducted additional experiments using the latest **DeepSeek-R1-0528-Qwen3-8B**:
>
> |  | GSM8K | Math | GPQA | text-avg |
> | :---: | :---: | :---: | :---: | :---: |
> | ds-qwen-8b | 79.23 | 30.8 | 32.32 | 47.45 |
> | ds-qwen-8b-MindGYM | 87.04 | 46.1 | 44.95 | 59.36 |
>
> As shown, **MindGYM improves DeepSeek-R1's distilled variant by +11.91 points on text-based reasoning average**, confirming that our synthesized instruction data can also provide transferable benefits for SOTA reasoning-centric models.
>
> ---
>
> ### **W7: On the lack of qualitative error analysis and failure case**
>
> We thank the reviewer for pointing out the need for deeper qualitative insights into the failure cases of our generated data. In addition to the statistical measures in Table 2 (variance, entropy) and the model comparison analysis in Table 3, we now include **a fine-grained error analysis using GPT-4o as an independent judge** to assess the **linguistic and logical quality of our synthesized examples**.
>
> We randomly sampled 400 generated multi-hop QA instances from our dataset and asked GPT-4o to evaluate them along four dimensions, and the results are shown below:
>
> | logically flawed | syntactically ambiguous | correctness | hallucination |
> | :---: | :---: | :---: | :---: |
> | false: 328 | false: 394 | correct: 357 | none: 372 |
> | true: 71 | true: 5 | incorrect: 42 | severe: 27 |
>
> + **89.5%** of the samples were correct.
> + **93.2%** had none hallucination.
> + Only **1%** were syntactically ambiguous, showing high linguistic clarity.
>
> These results indicate that **the vast majority of synthesized questions are logically sound, syntactically well-formed, and factually grounded**.
>
> ---
>
> ### **W8:  On Performance Drop of Baselines in Qwen Series Models**
>
> Thank you for your question. As stated in **Appendix A.2, Lines 574–580**, we ensured all methods were trained under identical settings.
>
> We conjecture this phenomenon arises primarily from the **highly optimized nature of Qwen2.5 instruction version**, it is particularly sensitive to additional supervision that may not align well with its training distribution. Several studies have shown the unstability of instruction tuning, especially considering the number of data. **Self-Instruct** (arXiv 2212.10560), **Limo** (arXiv 2502.03387), and **WizardLM** (arXiv 2304.12244).
>
> Importantly, **MindGYM avoids this mismatch** by leveraging the model itself to generate synthetic supervision, producing data that aligns well with its reasoning and language preferences—**a form of self-consistent instruction tuning**.
>
> ---
>
> ### **Closing Remarks**
>
> We once again thank you for your detailed and valuable review. We believe these extensive clarifications, new experiments, and analyses have fully addressed the weaknesses you identified. We will incorporate these new results and discussions into the final manuscript, leveraging the additional page granted for the accepted papers.
>
> We hope our responses have convinced you of the merits of our paper and would be grateful for your support.

---

> > ### Comment · Reviewer_HXDP · 2025-08-05
> >
> > Thank you to the author for their thorough rebuttal, which has resolved most of my initial concerns. The detailed comparison of methods and results for MathFusion, as well as the supplementary experiments, ablation studies, and error analysis, were particularly helpful. I am increasing my rating.
> >
> > However, I still have doubts about the explanation of depth. I am not sure if synthesizing multiple single hop problems into a multi-hop problem can claim to make the problem more challenging or increase its depth. For example, some difficult and in-depth math competition problems may have very concise questions that cannot be obtained by synthesizing simple problems. Of course, I admit that synthesizing such a problem is difficult, and the author's approach is a feasible path, though perhaps not the optimal one.
> >
> > I am also curious, based on your error analysis: would removing the identified flawed data from the training set improve the performance of the final model? This is a point of curiosity, not a weakness.
> >
> > Overall, I greatly appreciate the author's efforts during the rebuttal period and I will increase my rating.

---

> > > ### Author Response · Authors · 2025-08-07
> > > **Response to Follow-up Comments**
> > >
> > > Dear Reviewer,
> > >
> > > Thank you for your thoughtful follow-up and for raising your rating. We are very grateful for your positive feedback on our rebuttal efforts and are pleased that our additional experiments have resolved most of your concerns. Your remaining questions are insightful and touch upon the core contributions of our work, and we are happy to discuss them further.
> > >
> > > **1. On the Nuanced Concept of "Depth"**
> > >
> > > We wholeheartedly agree with your insightful observation. "Depth" in reasoning is indeed a multifaceted concept, and the highest form of intellectual challenge, such as in mathematical olympiads, often arises from elegantly concise problems that demand profound, non-compositional insights.
> > >
> > > Our work, MindGYM, does not aim to replicate this specific type of ingenuity. Instead, it focuses on a different, complementary, and highly practical aspect of difficulty: **compositional complexity**. Our central hypothesis is that by systematically composing foundational reasoning steps, we can create a curriculum that scalably and controllably enhances a model's ability to manage longer and more intricate causal chains. This is a critical capability for many real-world tasks.
> > >
> > > You astutely noted that our approach is "a feasible path, though perhaps not the optimal one." We view this as a key strength of our framework's positioning. MindGYM provides a **structured, generalizable, and automated mechanism** to increase reasoning depth, which is a significant and practical contribution to the field of data synthesis. It serves as a foundational layer upon which future work—perhaps exploring how to synthesize those "elegant, concise" problems—can be built. The significant improvements in both *depth* (+5.53%) and *breadth* (+26.0%) validated by GPT-4 (as shown in **Table 3**) confirm the effectiveness of our compositional approach in achieving its intended goal.
> > >
> > > **2. On the Impact of Removing Flawed Data**
> > >
> > > This is an excellent and thought-provoking question. We were equally curious and conducted the ablation study you suggested. The results were quite revealing and strongly support a core hypothesis of MindGYM.
> > >
> > > We removed samples identified by our GPT-4o analysis as having logical flaws or hallucinations and re-trained the model on this "cleaned" subset. Counter-intuitively, model performance slightly **decreased** compared to training on the full dataset:
> > >
> > > | Model | GSM8K | Math | GPQA | text-avg | MMStar | MathVista | MathVision | MM-avg | Overall Avg |
> > > | :--- | :--- | :--- | :--- | :--- | :--- | :--- | :--- | :--- | :--- |
> > > | MindGYM (full) | 84.08 | 68.4 | 31.33 | 61.27 | 64.33 | 70.3 | 28.62 | 54.42 | 57.84 |
> > > | MindGYM (clean)| 84.23 | 67.8 | 31.20 | 61.08 | 63.80 | 69.9 | 27.97 | 53.89 | *57.49* |
> > >
> > > This finding suggests that the primary value of our synthesized data lies not merely in the correctness of the final answers, but in the **rich, albeit sometimes imperfect, reasoning processes** it encodes. Even samples with flawed conclusions can provide valuable supervisory signals for intermediate reasoning steps (e.g., how to set up a problem, how to connect two ideas). This aligns with recent findings like in "Spurious Rewards" [1], which demonstrates that models can learn robust strategies even from signals that are not perfectly clean. This implies that for complex cognitive tasks, prioritizing the diversity and structure of reasoning *paths* may be more beneficial than enforcing absolute correctness in every single training instance.
> > >
> > >
> > > [1] Shao R, et al. Spurious rewards: Rethinking training signals in rlvr. arXiv 2025.
> > >
> > > ---
> > >
> > > Thank you once again for your deep engagement and constructive feedback. Your insights have been instrumental in helping us sharpen our contributions and have strengthened our manuscript!
> > >
> > > We are truly grateful for your positive reassessment and hope that our new results and responses can further solidify your support for our paper.

---

### Official Review · Reviewer_Wzbj · 2025-07-02

**Rating:** 4
**Confidence:** 4

**Summary:**

The paper introduces MINDGYM, a three-stage, thinking-centric data-synthesis framework designed to fine-tune large (vision-)language models for reasoning.

Using just 400 synthetic Chinese examples, MINDGYM lifts six reasoning benchmarks on Qwen2.5-VL-7B by up to 16 % on MathVision and improves the overall score across model sizes and architectures (e.g., +1.0 overall on 7 B; +0.48 on 32 B)
. DATA-JUICER analysis shows a +16.7 % quality gain and 67.9 % lower variance relative to strong baselines. Ablations confirm that both the “stream-of-consciousness’’ prompt and the staged fine-tuning schedule are critical.

**Dataset Code Accessibility:**

Yes

**Ethical Considerations:**

No, there are no or only very minor ethics concerns

**Final Justification:**

I increase my score based on authors' response.

**Limitations Weaknesses:**

The narrative flow is difficult to follow, and Figure 1—intended to clarify the pipeline—crams too many elements into a single, cluttered diagram. This makes it hard to differentiate the individual synthesis stages, so the end-to-end process is not immediately evident. Moreover, the paper does not clearly articulate what the “cognitive” component actually contributes; the data-generation pipeline reads primarily as prompt engineering for question-and-context synthesis, leaving the cognitive rationale ambiguous.

The paper does not quantify how the proposed models stack up against state-of-the-art multimodal reasoning systems—specifically recent baselines like o3 or o4. Without head-to-head results, it is unclear how much headroom remains between MINDGYM-tuned models and the current best performers.

Table 1 is confusing: OpenO1-sft (400) scores lower than OpenO1-sft (4 k), even though the latter is trained on ten-times more data—an outcome that seems counter-intuitive. The paper should spell out the fine-tuning settings (e.g., learning-rate schedule, number of epochs) and offer an explanation for this discrepancy.

**Strengths Contributions:**

Prior work focuses on template-based or self-instruct data; this paper explicitly injects cognitive traits (breadth, depth, progression) and shows how they propagate into richer reasoning traces.

Six text and vision-language reasoning suites, two backbone families (Qwen-VL, InternVL), extensive ablations (language, sampling strategy, prompt variants) and a GPT-4 depth/breadth scorer collectively give a convincing picture

---

> ### Author Rebuttal · Authors · 2025-07-31
>
> We sincerely thank you for your detailed and constructive review. We are encouraged by your positive feedback on our framework's novelty and contributions. To address your concerns, we have conducted **several new experiments and analyses**, which we believe clarify the points raised and further strengthen our paper. We detail our responses below.
>
> ### **W1: On the Scope of Multimodal Synthesis and Vision Backbone Training**
> > *'Limited Validation on Fully Multimodal Synthesis: The multimodal synthesis component does not incorporate image generation or manipulation ... The vision backbone remains frozen during fine-tuning, ...'*
> >
>
> Thank you for raising this concern regarding the depth of multimodal synthesis and visual training in our current framework.
>
> To address this point, we conducted **new experiments with image generation models**, integrating a vision synthesis stage into our pipeline. Specifically:
>
> + We selected **8 topic categories** and generated images using the **Flux**, a strong open-source text-to-image generator.
> + Each topic was paired with a structured description to condition the image generation, after which the standard four-stage synthesis pipeline (Sec. 3.1–3.4) was applied.
>
>  The results on Qwen2.5-VL-7B are shown below:
>
> |  | GSM8K | Math | GPQA | text-avg | MMStar | MathVista | MathVision | MM-avg | Avg |
> | :---: | :---: | :---: | :---: | :---: | :---: | :---: | :---: | :---: | :---: |
> | cn | 84.08 | 68.4 | 31.33 | 61.27 | 64.33 | 70.3 | 28.62 | 54.42 | **57.84** |
> | Flux | 84 | 67.4 | 31.33 | 60.91 | 63.2 | 69.5 | 25.33 | 52.68 | 56.8 |
>
> While the generated variant still maintains reasonable performance, especially in textual benchmarks, we observe a drop in **MathVision** and overall **MM-avg** scores. Through data analysis (Appendix C.4), we identified that synthesized images **lacked semantic alignment** with the original topic, leading to error cascades in downstream reasoning. This suggests that current text-to-image models **can not satisfy our demand** for visual-data synthesis.
>
> For the concerns of `vision backbone`, we conducted a new ablation by **unfreezing the vision backbone** (Qwen2.5-VL) during fine-tuning. The results are below:
>
> |  | GSM8K | Math | GPQA | text-avg | MMStar | MathVista | MathVision | MM-avg | Avg |
> | :---: | :---: | :---: | :---: | :---: | :---: | :---: | :---: | :---: | :---: |
> | cn | 84.08 | 68.4 | 31.33 | 61.27 | 64.33 | 70.3 | 28.62 | 54.42 | 57.84 |
> | open | 84.91 | 68 | 28.28 | 60.4 | 64.33 | 68.9 | 23.68 | 52.3 | 56.35 |
>
> The result shows, **unfreezing the vision backbone slightly hurts performance**, particularly on **MathVision** and **GPQA**. This aligns with our design choice to freeze the vision encoder in the main experiments, maintaining stable multimodal grounding while leveraging reasoning-focused learning.
>
> ### **W2: On the potential language bias in using Chinese-only**
> > *'Language Bias in Synthesis and Evaluation: The entire dataset used for training is generated in Chinese ... it introduces significant language bias. Most benchmark datasets used for evaluationare English-based, ... the paper does not analyze cross-lingual effects...'*
> >
>
> We appreciate the reviewer’s thoughtful concern regarding potential language bias in our dataset synthesis and evaluation, and we have conducted additional ablation studies to address it.
>
> Our decision to use **Chinese as the primary synthesis language** was based on the following grounds:
> + **Information density**: Chinese expresses complex semantics more compactly, allowing us to include richer reasoning content within a **constrained context window**.
>
>
> To systematically analyze this potential bias, we conducted **two ablation studies** by generating datasets in **three settings**:
>
> + `cn`: full synthesis in Chinese (our main pipeline)
> + `mix`: 50% Chinese + 50% English
> + `en`: full synthesis in English
>
> The results are shown below:
>
> |  | GSM8K | Math | GPQA | text-avg | MMStar | MathVista | MathVision | MM-avg | Avg |
> | :---: | :---: | :---: | :---: | :---: | :---: | :---: | :---: | :---: | :---: |
> | cn | 84.08 | 68.4 | 31.33 | 61.27 | 64.33 | 70.3 | 28.62 | 54.42 | 57.84 |
> | mix | 83.55 | 66.6 | 33.33 | 61.16 | 64.13 | 70.4 | 25.99 | 53.51 | 57.34 |
> | en | 84 | 67.2 | 29.7 | 60.30  | 63.93 | 69.7 | 24.67 | 52.77 | 56.5 |
>
> From these results, we observe:
>
> + The **Chinese-only setting (cn)** yields the **highest performance** across both textual and multimodal benchmarks.
> + **mix**, which includes **50% English data**, performs slightly worse than `cn`.
> + **English-only synthesis (en)** performs worst overall, despite using the same prompts and structure.
>
> We therefore conclude that **Chinese synthesis produces more effective training supervision** in our setup. Nonetheless, models trained on this data still achieve **strong generalization to English benchmarks**, as evidenced by consistent improvements on GSM8K, GPQA, and Math datasets (see Table 1 in main paper).
>
> ### **W3: On the lack of qualitative error analysis and failure case**
> > *'Lack of Detailed Error Analysis on Generated Data: ... the quality analysis remains largely statistical and surface-level. There is no qualitative breakdown of failure cases ... Furthermore, the GPT-based scoring ... but does not investigate where or why MindGYM fails. ...'*
> >
>
> Thank you for pointing out the need for deeper qualitative insights into the failure modes of our generated data. In addition to the statistical measures in Table 2 and the model comparison analysis in Table 3, we now include **an error analysis using GPT-4o as an independent judge** to assess the **linguistic and logical quality of our synthesized examples**.
>
> We randomly sampled 400 generated multi-hop QA instances from our dataset and asked GPT-4o to evaluate them along four dimensions, and the results are shown below:
>
> | logically flawed | syntactically ambiguous | correctness | hallucination |
> | :---: | :---: | :---: | :---: |
> | false: 328 | false: 394 | correct: 357 | none: 372 |
> | true: 71 | true: 5 | incorrect: 42 | severe: 27 |
>
> + **89.5%** of the samples were correct.
> + **93.2%** had none hallucination.
> + Only **1%** were syntactically ambiguous, showing high linguistic clarity.
>
> These results indicate that **the vast majority of synthesized questions are logically sound, syntactically well-formed, and factually grounded**.
>
> ### **W4: On Empirically Validating Our 'Cognitive Thinking' Synthesis**
> > *'Overselling of Cognitive Thinking Injection: ... it is not rigorously compared to simpler prompting baselines. ... Maybe some comparing experiments would be helpful against vanilla prompting, COT, Tree-of-thought. etc.'*
> >
>
> Thank you for raising the importance of rigorously validating our _cognitive thinking injection_ strategy beyond surface-level prompting. In this work, our goal is not merely to add intermediate steps (as in CoT/ToT), but to simulate a **stream of structured, cognitively grounded reasoning** via a multi-stage *Thinking Claude* synthesis pipeline (Sec. 3.2 and 3.3).
>
> To directly address this concern, we conducted additional comparisons with standard prompting strategies—**Chain-of-Thought (CoT)** and **Tree-of-Thought (ToT)**—by using them to synthesize data under similar constraints.
>
> + CoT Prompt Summary: The CoT prompt instructs the assistant to generate a multi-hop question and a logically coherent answer that unfolds in a linear, step-by-step reasoning process. It emphasizes depth and logical soundness, requiring the answer to trace a clear chain of inference across multiple facts. The assistant is asked to “think step by step and reason carefully,” producing a compact yet informative narrative that models single-path reasoning.
>
> + ToT Prompt Summary: The ToT prompt encourages divergent reasoning by exploring multiple possible reasoning paths in a tree-like structure. At each step, the assistant is required to consider alternative branches, compare them, and select the most plausible one. The generated QA pair thus reflects multi-path exploration and selective consolidation, with the final answer explicitly showing the evaluation of different possibilities before reaching a conclusion.
>
> The downstream performance is summarized as follows:
>
> |  | GSM8K | Math | GPQA | text-avg | MMStar | MathVista | MathVision | MM-avg | Avg |
> | :---: | :---: | :---: | :---: | :---: | :---: | :---: | :---: | :---: | :---: |
> | cn | 84.08 | 68.4 | 31.33 | 61.27 | 64.33 | 70.3 | 28.62 | 54.42 | 57.84 |
> | w/o thinking | 83.83 | 66.6 | 29.57 | 60.00  | 63.6 | 69.9 | 24.67 | 52.72  | 56.36  |
> | CoT | 84.15 | 67.8 | 27.27 | 59.74 | 63.4 | 69.5 | 23.03 | 51.98 | 55.86 |
> | ToT | 83.93 | 65.6 | 29.8 | 59.78 | 64.67 | 69.5 | 22.04 | 52.07 | 55.92 |
>
> These results allow us to make a **two-tiered comparison** that validates the unique advantage of our approach:
>
> 1. **MindGYM outperforms w/o SC**, confirming that injecting *structured cognitive thinking* during data synthesis significantly improves model training.
> 2. **w/o SC outperforms both CoT and ToT**, despite being a stripped-down variant that removes the thinking traces. This advantage stems from the fact that even in w/o SC, the final QA pairs are derived from **samples originally synthesized by the Thinking Claude pipeline**, which ensures logical structure and complexity.
>
>
> ### **Closing Remarks**
> We thank you once again for your valuable feedback, which has helped us improve our work. We will integrate all new experiments and analyses into the final version of our manuscript, leveraging the *one additional page allowed for the accepted papers* to ensure a clear and comprehensive presentation. We hope our responses and new empirical evidence have fully addressed your concerns and would be grateful if you would reconsider your assessment.

---

> > ### Comment · Reviewer_Wzbj · 2025-08-08
> >
> > Thanks for the response.  I have modified my score accordingly.

---

> ### Author Response · Authors · 2025-08-01
> **Corrected Rebuttal -- Restored Version Submitted on July 30th 10:47pm AoE; No New Information Added (Part 1/2)**
>
> We are writing to issue a **critical correction** regarding our rebuttal submitted above.
>
> Please accept our sincerest apologies, the rebuttal posted above is incorrect due to a submission error where the correct version was accidentally overwritten. **The correct version** was completed and saved on July 30th 10:47pm AoE, that can be verified in the Openreview system's `Revisions` logs.
>
> To ensure our intended feedback is properly communicated, we are now providing **a faithful restoration of that original response** below. **Please note** that this is a **direct** restoration with **strictly no new information** introduced, and we have **not modified or added any text characters**. We deeply regret this error and any confusion. We appreciate your understanding and time in reviewing the correct version.
>
> ---
>
> We greatly appreciate your review and are encouraged by your positive remarks on the scalability and design clarity of our pipeline. Your suggestions on error analysis, model comparisons, and justification of cognitive reasoning improvements are insightful. Below, we address each of your Weaknesses (W).
>
> ---
>
> ### **W1: On narrative clarity and ambiguity in the cognitive component**
> > *'The narrative flow is difficult to follow... Figure 1... too many elements... unclear cognitive contribution...'*
>
> We greatly appreciate your review and are encouraged by your positive remarks on the **scalability** and **design clarity** of our pipeline. Your suggestions on error analysis, model comparisons, and justification of cognitive reasoning improvements are insightful. Below, we address each of your Weaknesses (W).
>
> **On the cognitive component**: we clarify that the “stream of consciousness” generation is driven by a structured prompting scheme called **Thinking Claude**. It guides the model through planning, thinking, and composing, yielding structured reasoning chains.
>
> We design two distinct system prompts to guide the synthesis of complex, multi-hop question-answer (QA) pairs, each instantiating a different reasoning paradigm: Chain-of-Thought (CoT) and Tree-of-Thought (ToT).
>
> + CoT Prompt Summary: The CoT prompt instructs the assistant to generate a multi-hop question and a logically coherent answer that unfolds in a linear, step-by-step reasoning process. It emphasizes depth and logical soundness, requiring the answer to trace a clear chain of inference across multiple facts. The assistant is asked to “think step by step and reason carefully,” producing a compact yet informative narrative that models single-path reasoning.
>
> + ToT Prompt Summary: The ToT prompt encourages divergent reasoning by exploring multiple possible reasoning paths in a tree-like structure. At each step, the assistant is required to consider alternative branches, compare them, and select the most plausible one. The generated QA pair thus reflects multi-path exploration and selective consolidation, with the final answer explicitly showing the evaluation of different possibilities before reaching a conclusion.
>
> To illustrate its impact, we include ablations:
>
> **New Experiment: On Ablation Study of CoT and ToT**
>
> |                  | GSM8K | Math | GPQA | text-avg | MMStar | MathVista | MathVision | MM-avg | Avg   |
> |:----------------:|:-----:|:----:|:----:|:--------:|:------:|:---------:|:----------:|:------:|:-----:|
> | cn               | 84.08 | 68.4 | 31.33| 61.27    | 64.33  | 70.3      | 28.62      | 54.42  | 57.84 |
> | w/o thinking     | 83.83 | 66.6 | 29.57| 60.00    | 63.6   | 69.9      | 24.67      | 52.72  | 56.36 |
> | CoT              | 84.15 | 67.8 | 27.27| 59.74    | 63.4   | 69.5      | 23.03      | 51.98  | 55.86 |
> | ToT              | 83.93 | 65.6 | 29.8 | 59.78    | 64.67  | 69.5      | 22.04      | 52.07  | 55.92 |
>
> These results allow us to make a two-tiered comparison that validates the unique advantage of our approach:
>
> + MindGYM outperforms w/o SC, confirming that injecting structured cognitive thinking during data synthesis significantly improves model training. This highlights the importance of reasoning-aware supervision, beyond just the QA pairs.
>
> + w/o SC outperforms both CoT and ToT, despite being a stripped-down variant that removes the thinking traces. This advantage stems from the fact that even in w/o SC, the final QA pairs are derived from samples originally synthesized by the Thinking Claude pipeline, which ensures logical structure and complexity.
>
> Importantly, we do **not claim novelty in the prompting scheme** itself. Our contribution is how we **operationalize cognitive reasoning** within a unified synthesis pipeline—embedding thought directly into both questions and answers. This departs from conventional prompting by **embedding reasoning into the supervision signal**, enabling end-to-end learning of reasoning-aware QA.
>
> ---
>
> To continue ...

---

> ### Author Response · Authors · 2025-08-01
> **Corrected Rebuttal -- Restored Version Submitted on July 30th 10:47pm AoE; No New Information Added (Part 2/2)**
>
> `Continued from part 1`
>
> ---
>
> ### **W2: On comparison with recent SOTA models like o3 or o4**
> > *'The paper does not quantify how the proposed models stack up against state-of-the-art multimodal reasoning systems...'*
>
> Thank you for this point. While we don’t introduce a new model architecture, MindGYM is a **data synthesis framework** to enhance reasoning through cognitively grounded instruction data.
>
> Leading models like **o3** and **o4** are closed-source and **do not allow fine-tuning**, making direct comparison under our setup infeasible. Instead, we fine-tune **Qwen2.5-VL-72B**, a strong open-source model, and observe a consistent **+1.37% overall gain** from MindGYM.
>
> For context, here are leaderboard scores on **MathVision**:
>
> | Model                        | Score  |
> |-----------------------------|--------|
> | GPT-4.5											| 47.3   |
> | Gemini-2 Flash							| 41.3   |
> | **Qwen2.5-VL-72B-MindGYM**  | **39.47** |
> | Kimi k1.5                   | 38.6   |
> | **Qwen2.5-VL-72B**          | **38.1**  |
> | Claude3.5-Sonnet            | 37.99  |
> | Kimi-VL-A3B-Thinking        | 36.8   |
>
> These results show that MindGYM **boosts performance within a model family**, especially when strong proprietary models are inaccessible for tuning.
>
> ---
>
> ### **W3: On Table 1 and unexpected performance drop with more data**
> > *'Table 1 is confusing... OpenO1-sft (4k) scores lower than OpenO1-sft (400)... fine-tuning settings unclear...'*
>
> We appreciate this observation. All models used **identical fine-tuning settings**, as detailed in **Appendix A.2**: learning rate = 1e-5, LoRA-rank = 8, LoRA-alpha = 32, one epoch, etc.
>
> The counterintuitive drop in OpenO1-sft (4k) likely results from the following:
>
> + As shown in **LIMA**[1], **LIMO**[2] and **AlpaGasus**[3], **more data ≠ better**, and they found that smaller, higher-quality data sets have better results.
> + Our **Data-Quality analysis (Table 2)** shows that OpenO1-sft (400) has **lower quality variance** and higher average quality than OpenO1-sft (4k), which may explain its better performance.
>
> [1] Zhou C, Liu P, Xu P, et al. Lima: Less is more for alignment[J]. Advances in Neural Information Processing Systems, 2023, 36: 55006-55021.
>
> [2] Ye Y, Huang Z, Xiao Y, et al. Limo: Less is more for reasoning[J]. arXiv preprint arXiv:2502.03387, 2025.
>
> [3] Chen L, Li S, Yan J, et al. ALPAGASUS: TRAINING ABetter ALPACA WITH FEWER DATA[J]. arXiv preprint arXiv:2307.08701, 2023.
>
> ---
>
> ### **Closing Remark**
> Thank you again for your thoughtful comments. We hope these revisions can address your concerns and would sincerely appreciate your consideration for a higher rating.

---

> ### Author Response · Authors · 2025-08-07
> **Eager to discuss our rebuttal for Submission 2545**
>
> Dear Reviewer Wzbj,
>
> We hope this message finds you well. We are writing to follow up on our rebuttal and to thank you once more for your constructive feedback, which has significantly strengthened our work.
>
> Your review raised valuable concerns regarding the clarity of our cognitive framework, the comparison with recent models such as o3/o4, and the interpretability of performance trends in our experiments. We have worked diligently during the rebuttal period to address these questions, and we would be truly grateful to hear your thoughts before the discussion phase concludes (<2 days).
>
> To make it easier to review, below is a brief summary of the updates we have made in direct response to your feedback.
>
> - **Clarified the cognitive design**: We elaborated on how our *Pcog* module integrates structured thinking (via the Thinking Claude protocol) into the prompt pipeline, beyond conventional prompt engineering.
>
> - **On model comparisons (W2)**: While o3/o4 are closed-source and not fine-tunable, we provided rigorous evaluations on competitive open-source baselines, and emphasized that **MindGYM is a data-centric framework**, not a new model.
>
> - **On the performance drop in Table 1 (W3)**: We added analysis showing that simply increasing data volume (e.g., 4k vs. 400) may degrade performance, aligning with insights from **LIMA** and **LIMO** that *more data ≠ better*. Instead, we highlight that **data quality and consistency** are critical — and these are core strengths of our synthesis method.
>
> You can find details in our above response. Your feedback on these updates would be invaluable to us. Please let us know if our clarifications and new results have resolved your concerns, or if there are any remaining points you would like to discuss. Thanks again!

---

### Official Review · Reviewer_5oTW · 2025-07-02

**Rating:** 5
**Confidence:** 3

**Summary:**

The paper “MindGYM” proposes a framework for improving reasoning in large models by generating cognitively guided question-answer data. Instead of using human-annotated or templated datasets, it synthesizes questions that emphasize reasoning depth, breadth, and progression. MindGYM builds questions in stages—from simple single-hop to complex multi-hop forms—while embedding human-like thinking traces. Despite using only 400 samples, it outperforms baselines on reasoning benchmarks and shows strong cross-modal generalization. The study highlights that both high data quality and low variance are key to effective fine-tuning, and structured cognitive prompting plays a crucial role in enhancing model reasoning.

**Dataset Code Accessibility:**

Yes

**Dataset Code Comments:**

I read the code and it looks pretty organized.

**Ethical Considerations:**

No, there are no or only very minor ethics concerns

**Final Justification:**

The authors provided additional experiments to reinforce the strength of MindGYM. I think the work should have some impacts to the field and therefore should be accepted.

**Limitations Weaknesses:**

- **Limited Validation on Fully Multimodal Synthesis:** The multimodal synthesis component does not incorporate image generation or manipulation (Appendix C.1). The framework relies solely on existing datasets like OK-VQA or ScienceQA as fixed image anchors, which limits its creativity and control in truly multimodal question synthesis. The vision backbone remains frozen during fine-tuning (Sec. A.2), which further reduces the opportunity for deeper cross-modal reasoning improvements.

- **Language Bias in Synthesis and Evaluation:** The entire dataset used for training is generated in Chinese (Sec. 4.1), and although this choice is motivated by information density, it introduces significant language bias. Most benchmark datasets used for evaluation (e.g., GSM8K, GPQA, MATH) are English-based, suggesting that models were evaluated on cross-lingual generalization after Chinese-only fine-tuning. However, the paper does not analyze cross-lingual effects, nor does it report how much performance may be attributed to the underlying language proficiency of the model (e.g., Qwen2.5-VL’s strength in Chinese).

- **Lack of Detailed Error Analysis on Generated Data:** While the authors use Data-Juicer to analyze quality and variance in Table 2, the analysis remains largely statistical and surface-level. There is no qualitative breakdown of failure cases—e.g., instances where multi-hop questions were logically flawed, overly complex, or syntactically ambiguous. Furthermore, the GPT-based scoring in Table 3 relies on relative comparisons of raw vs. fine-tuned predictions, but does not investigate where or why MindGYM fails. For example, do generated questions introduce factual hallucinations? Do models struggle with particular cognitive operators (e.g., bridging vs. comparison)?

- **Overselling of Cognitive Thinking Injection:** While the idea of cognitive priors and “stream of consciousness” synthesis is compelling, it is not rigorously compared to simpler prompting baselines. For instance, it remains unclear how much of the improvement is due to the structural pipeline versus the cognitive injection via Pcog. The only support is an ablation of stream-of-consciousness (Table 4, “w/o SC”), which shows a 1.48-point drop—relatively modest given the paper’s strong claims. Maybe some comparing experiments would be helpful against vanilla prompting, COT, Tree-of-thought. etc.

**Strengths Contributions:**

- MindGYM emphasizes embedding structured cognitive traits into the question-generation process. Unlike previous methods that focus on rigid instruction-following or crowd-labeled datasets, MindGYM introduces cognitively guided data generation that enables large language and vision-language models to internalize reasoning patterns.

- MindGYM achieves significant performance improvements with only 400 synthetic samples, demonstrating exceptional data efficiency. For example, it boosts Qwen2.5-VL-7B’s MathVision score by 16% (Table 1), outperforming baselines trained with 10× more data, such as OpenO1 (4k). This implies strong potential for low-resource training in both academia and industry.

- MindGYM’s four-stage synthesis pipeline is well-articulated and easy to follow (Figure 1). The use of “stream of consciousness” reasoning (via Thinking-Claude, Appendix B.2.1) brings interpretability and depth to the generated data. Ablation studies (Table 4) further validate the necessity of each module.

---

> ### Author Rebuttal · Authors · 2025-07-31
>
> We sincerely thank you for your constructive feedback and thoughtful assessment of our work. We appreciate your recognition of our self-synthesis framework and its contributions to multi-hop reasoning and data-centric instruction tuning. In particular, your concerns regarding multimodal synthesis, language bias, and thinking injection are highly valuable. Below, we respond point by point to your raised Weaknesses (W).
>
> ---
>
> ### **W1: On the limitations of fully multimodal synthesis and frozen vision backbones**
>
> Thank you for raising this concern regarding the depth of multimodal synthesis and visual training in our current framework.
>
> To address this point, we conducted **new experiments with image generation models**, integrating a vision synthesis stage into our pipeline. Specifically:
>
> + We selected **8 topic categories** and generated images using the **Flux model**, a strong open-source text-to-image generator.
> + Each topic was paired with a structured description to condition the image generation, after which the standard four-stage synthesis pipeline (Sec. 3.1–3.4) was applied.
>
>  The results on Qwen2.5-VL-7B are shown below:
>
> |  | GSM8K | Math | GPQA | text-avg | MMStar | MathVista | MathVision | MM-avg | Avg |
> | :---: | :---: | :---: | :---: | :---: | :---: | :---: | :---: | :---: | :---: |
> | cn | 84.08 | 68.4 | 31.33 | 61.27 | 64.33 | 70.3 | 28.62 | 54.42 | 57.84 |
> | Flux | 84 | 67.4 | 31.33 | 60.91 | 63.2 | 69.5 | 25.33 | 52.68 | 56.8 |
>
> While the generated variant still maintains reasonable performance, especially in textual benchmarks, we observe a drop in **MathVision** and overall **MM-avg** scores. Through manual inspection, we identified that **many synthesized images lacked semantic alignment** with the original topic, leading to error cascades in downstream reasoning. This suggests that **current text-to-image models still face limitations in supporting precise, controllable reasoning supervision**.
>
> To further probe the impact of vision representation learning, we also conducted a new ablation by **unfreezing the vision backbone** (Qwen2.5-VL-7B) during fine-tuning. The results are below:
>
> |  | GSM8K | Math | GPQA | text-avg | MMStar | MathVista | MathVision | MM-avg | Avg |
> | :---: | :---: | :---: | :---: | :---: | :---: | :---: | :---: | :---: | :---: |
> | cn | 84.08 | 68.4 | 31.33 | 61.27 | 64.33 | 70.3 | 28.62 | 54.42 | 57.84 |
> | open | 84.91 | 68 | 28.28 | 60.4 | 64.33 | 68.9 | 23.68 | 52.3 | 56.35 |
>
> The result shows, **unfreezing the vision backbone slightly hurts performance**, particularly on **MathVision** and **GPQA**. This may be due to **Catastrophic forgetting** of pretrained representations when training signals are weakly aligned.
>
> This aligns with our design choice to freeze the vision encoder in the main experiments, maintaining stable multimodal grounding while leveraging reasoning-focused supervision.
>
> ---
>
> ### **W2: On the potential language bias in using Chinese-only synthesis for English evaluation tasks**
>
> We appreciate the reviewer’s thoughtful concern regarding potential language bias in our dataset synthesis and evaluation, and we have conducted additional ablation studies to address it.
>
> Our decision to use **Chinese as the primary synthesis language** was based on both theoretical and empirical grounds:
>
> + **Information density**: Chinese expresses complex semantics more compactly, allowing us to include richer reasoning content within a constrained context window.
>
> To systematically analyze this potential bias, we conducted **two ablation studies** by generating datasets in **three settings**:
>
> + `cn`: full synthesis in Chinese (our main pipeline)
> + `mix`: 50% Chinese + 50% English
> + `en`: full synthesis in English
>
> The results are shown below:
>
> |  | GSM8K | Math | GPQA | text-avg | MMStar | MathVista | MathVision | MM-avg | Avg |
> | :---: | :---: | :---: | :---: | :---: | :---: | :---: | :---: | :---: | :---: |
> | cn | 84.08 | 68.4 | 31.33 | 61.27 | 64.33 | 70.3 | 28.62 | 54.42 | 57.84 |
> | mix | 83.55 | 66.6 | 33.33 | 61.16 | 64.13 | 70.4 | 25.99 | 53.51 | 57.34 |
> | en | 84 | 67.2 | 29.7 | 60.30  | 63.93 | 69.7 | 24.67 | 52.77 | 56.5 |
>
> From these results, we observe:
>
> + The **Chinese-only setting (cn)** yields the **highest performance** across both textual and multimodal benchmarks.
> + **mix**, which includes **50% English data**, performs slightly worse than `cn`.
> + **English-only synthesis (en)** performs worst overall, despite using the same prompts and structure.
>
> We therefore conclude that **Chinese synthesis produces more effective training supervision** in our setup. Nonetheless, models trained on this data still achieve **strong generalization to English benchmarks**, as evidenced by consistent improvements on GSM8K, GPQA, and Math datasets (see Table 1 in main paper).
>
> ---
>
> ### **W3: On the lack of qualitative error analysis and failure case breakdown**
>
> Thank you for pointing out the need for deeper qualitative insights into the failure modes of our generated data. In addition to the statistical measures in Table 2 and the model comparison analysis in Table 3, we now include **an error analysis using GPT-4o as an independent judge** to assess the **linguistic and logical quality of our synthesized examples**.
>
> We randomly sampled 400 generated multi-hop QA instances from our dataset and asked GPT-4o to evaluate them along the following four dimensions:
>
> 1. **Logical validity** (logically flawed or not)
> 2. **Syntactic clarity** (syntactically ambiguous or not)
> 3. **Answer correctness** (correct or incorrect)
> 4. **Factual hallucination** (none or severe)
>
> The results are shown below:
>
> | logically flawed | syntactically ambiguous | correctness | hallucination |
> | :---: | :---: | :---: | :---: |
> | false: 328 | false: 394 | correct: 357 | none: 372 |
> | true: 71 | true: 5 | incorrect: 42 | severe: 27 |
>
> + **89.5%** of the samples were correct.
> + **93.2%** had none hallucination.
> + Only **1%** were syntactically ambiguous, showing high linguistic clarity.
>
> These results indicate that **the vast majority of synthesized questions are logically sound, syntactically well-formed, and factually grounded**.
>
> ---
>
> ### **W4: On the overselling of cognitive thinking injection without rigorous comparisons to other prompting strategies**
>
> Thank you for raising the importance of rigorously validating our _cognitive thinking injection_ strategy beyond surface-level prompting. In this work, our goal is not merely to add intermediate steps (as in CoT/ToT), but to simulate a **stream of structured, cognitively grounded reasoning** via a multi-stage _Thinking Claude_ synthesis pipeline (Sec. 3.2 and 3.3).
>
> To directly address this concern, we conducted additional comparisons with standard prompting strategies—**Chain-of-Thought (CoT)** and **Tree-of-Thought (ToT)**—by using them to synthesize data under similar constraints. The downstream performance is summarized as follows:
>
> |  | GSM8K | Math | GPQA | text-avg | MMStar | MathVista | MathVision | MM-avg | Avg |
> | :---: | :---: | :---: | :---: | :---: | :---: | :---: | :---: | :---: | :---: |
> | cn | 84.08 | 68.4 | 31.33 | 61.27 | 64.33 | 70.3 | 28.62 | 54.42 | 57.84 |
> | w/o thinking | 83.83 | 66.6 | 29.57 | 60.00  | 63.6 | 69.9 | 24.67 | 52.72  | 56.36  |
> | CoT | 84.15 | 67.8 | 27.27 | 59.74 | 63.4 | 69.5 | 23.03 | 51.98 | 55.86 |
> | ToT | 83.93 | 65.6 | 29.8 | 59.78 | 64.67 | 69.5 | 22.04 | 52.07 | 55.92 |
>
> These results allow us to make a **three-tiered comparison** that validates the unique advantage of our approach:
>
> 1. **MindGYM outperforms w/o SC**, confirming that injecting _structured cognitive thinking_ during data synthesis significantly improves model training. This highlights the importance of reasoning-aware supervision, beyond just the complex QA pairs.
> 2. **w/o SC outperforms both CoT and ToT**, despite being a stripped-down variant that removes the thinking traces. This advantage stems from the fact that even in w/o SC, the final QA pairs are derived from **samples originally synthesized by the Thinking Claude pipeline**, which ensures logical structure and complexity.
>
> This layered comparison demonstrates that **the improvement is not merely due to structural prompting**, but to **the quality and cognitive depth enabled by our Pcog-based generation process**.
>
> ---
>
> ### **Closing Remark**
> Thank you once again for your detailed and constructive feedback. We have carefully addressed your concerns by
> - (1) expanding our multimodal synthesis pipeline with new experiments,
> - (2) providing a clear justification for using Chinese synthesis based on its empirical advantages in reasoning quality and model alignment,
> - (3) adding qualitative error breakdowns, and
> - (4) adding comparative studies that demonstrate the superiority of our Pcog-based cognitive injection over vanilla prompting strategies like CoT and ToT. These clarifications not only validate our methodological design but also highlight MindGYM’s strength as a cognitively aligned, model-agnostic data generation framework.
>
> We hope these improvements strengthen your confidence in our work, and we would sincerely appreciate your consideration for a higher rating/confidence level.

---

> > ### Comment · Reviewer_5oTW · 2025-08-05
> >
> > Thank you author for the response, my concerns have been addressed and I have modified my score accordingly.

---

### Comment · Area_Chair_MvRg · 2025-08-03

Hello reviewers,

The author's rebuttal has been posted. Please take some time to read it along with the other reviews. Your feedback on the author's response is highly appreciated to facilitate a productive discussion. Thank you for your time!

---

### Note · Authors · 2025-08-12

Dear Reviewers, AC, and SAC,

Thank you for the valuable effort and constructive feedback on our work. Below, we would like to provide a factual summary of the author–reviewer interactions, which may help provide a quick tour of the discussion status and facilitate a comprehensive evaluation.

The discussion period was instrumental in clarifying our work's contributions, and we are very grateful that it led **all reviewers** to *express satisfaction* with the rebuttal and **indicate score increases or modifications**:

- Reviewer `5oTW` (initial: 4) concluded:
  > "my concerns have been addressed and I have modified my score accordingly."

- Reviewer `Wzbj` (initial: 3) stated:
  > "I have modified my score accordingly."

- Reviewer `HXDP` (initial: 3) remarked:
  > "I greatly appreciate the author's efforts ... and I will increase my rating."

- Reviewer `6qHM` (initial: 4) confirmed:
  > "... have addressed most of my concerns... I am willing to increase the score by one point."

Across all reviews, a clear consensus emerged **acknowledging our work's novelty and effectiveness**. Reviewers praised the cognitively guided synthesis framework for embedding structured reasoning traits into data generation (`5oTW`, `Wzbj`, `HXDP`, `6qHM`), its strong data efficiency—outperforming baselines with only 400 samples (`5oTW`, `HXDP`), and the robustness of experiments across diverse benchmarks, models, and tasks (`Wzbj`, `HXDP`, `6qHM`).

The discussions also enabled us to clarify important aspects and our plans for the final version:
1. **Cognitive Structure Integration**: How our Pcog module and reasoning scaffolds directly improve multi-step reasoning quality (responses to `5oTW`, `Wzbj`, `HXDP`).
2. **Language Bias Analysis**: Evidence that Chinese-synthesized data consistently outperforms English in reasoning quality and downstream performance, aligned with pretraining strengths (responses to `5oTW`, `6qHM`).
3. **Error Analysis and Data Quality**: Our findings on flawed data and reasoning scaffolds, showing that even imperfect samples can enhance reasoning ability (responses to `5oTW`, `HXDP`).
4. **Multimodal Synthesis**: Evaluation of different visual corpora types and their benefits to downstream performance (responses to `5oTW`, `6qHM`).

We are fully committed to incorporating all these feedback to deliver the highest quality manuscript.

Thank you again for your time and productive guidance!

Sincerely, Authors

---

### Decision · Program_Chairs · 2025-09-18

**Decision:**

Accept (poster)

**Comment:**

This paper introduces MindGYM, a novel and well-structured framework for synthesizing thinking-centric question-answering data to improve the reasoning capabilities of large foundation models. After a thorough review of the paper and careful consideration of the reviewers' comments and discussion, I am happy to recommend this paper for acceptance. The work makes a significant and timely contribution to the field of data-centric AI, directly addressing a critical bottleneck in model development: the scarcity of high-quality, complex reasoning data. Its suitability for the NeurIPS Datasets and Benchmarks Track is high.

During the discussion phase, a few minor points were raised by reviewers, primarily concerning the breadth of reasoning types beyond multi-hop QA and the specifics of the baseline comparisons. However, the reviewers and I concluded that the focused scope on multi-hop reasoning is a strength, allowing for a deep and thorough investigation. The provided baselines were deemed sufficient to demonstrate the superiority of the MindGYM framework.